# Beyond Homophily in Graph Neural Networks: Current Limitations and Effective Designs

**Jiong Zhu**
University of Michigan
jiongzhu@umich.edu

**Yujun Yan**
University of Michigan
yujunyan@umich.edu

**Lingxiao Zhao**
Carnegie Mellon University
lingxia1@andrew.cmu.edu

**Mark Heimann**
University of Michigan
mheimann@umich.edu

**Leman Akoglu**
Carnegie Mellon University
lakoglu@andrew.cmu.edu

**Danai Koutra**
University of Michigan
dkoutra@umich.edu

## Abstract

We investigate the representation power of graph neural networks in the semi-supervised node classification task under *heterophily* or *low homophily*, i.e., in networks where connected nodes may have *different* class labels and *dissimilar* features. Many popular GNNs fail to generalize to this setting, and are even outperformed by models that ignore the graph structure (e.g., multilayer perceptrons). Motivated by this limitation, we identify a set of key designs—ego- and neighbor-embedding separation, higher-order neighborhoods, and combination of intermediate representations—that boost learning from the graph structure under heterophily. We combine them into a graph neural network, $H_2$GCN, which we use as the base method to empirically evaluate the effectiveness of the identified designs. Going beyond the traditional benchmarks with strong homophily, our empirical analysis shows that the identified designs increase the accuracy of GNNs by up to 40% and 27% over models without them on synthetic and real networks with heterophily, respectively, and yield competitive performance under homophily.

## 1 Introduction

We focus on the effectiveness of graph neural networks (GNNs) [42] in tackling the semi-supervised node classification task in challenging settings: the goal of the task is to infer the unknown labels of the nodes by using the network structure [44], given partially labeled networks with node features (or attributes). Unlike most prior work that considers networks with strong homophily, we study the representation power of GNNs in settings with *different levels of homophily* or *class label smoothness*.

Homophily is a key principle of many real-world networks, whereby linked nodes often belong to the same class or have similar features ("birds of a feather flock together") [21]. For example, friends are likely to have similar political beliefs or age, and papers tend to cite papers from the same research area [23]. GNNs *model the homophily principle* by propagating features and aggregating them within various graph neighborhoods via different mechanisms (e.g., averaging, LSTM) [17, 11, 36]. However, in the real world, there are also settings where "opposites attract", leading to networks with *heterophily*: linked nodes are likely from different classes or have dissimilar features. For instance, the majority of people tend to connect with people of the opposite gender in dating networks, different amino acid types are more likely to connect in protein structures, fraudsters are more likely to connect to accomplices than to other fraudsters in online purchasing networks [24].

Since many existing GNNs assume strong homophily, they fail to generalize to networks with heterophily (or low/medium level of homophily). In such cases, we find that even models that ignore

the graph structure altogether, such as multilayer perceptrons or MLPs, can outperform a number of existing GNNs. Motivated by this limitation, we make the following contributions:

- **Current Limitations**: We reveal the limitation of GNNs to learn over networks with heterophily, which is ignored in the literature due to evaluation on few benchmarks with similar properties. § 3
- **Key Designs for Heterophily & New Model:** We identify a set of key designs that can boost learning from the graph structure in heterophily without trading off accuracy in homophily: (D1) ego- and neighbor-embedding separation, (D2) higher-order neighborhoods, and (D3) combination of intermediate representations. We justify the designs theoretically, and combine them into a model, $H_2$GCN, that effectively adapts to both heterophily and homophily. We compare it to prior GNN models, and make our code and data available at `https://github.com/GemsLab/H2GCN`. § 3-4
- **Extensive Empirical Evaluation**: We empirically analyze our model and competitive existing GNN models on both synthetic and real networks covering the full spectrum of low-to-high homophily (besides the typically-used benchmarks with strong homophily only). In synthetic networks, our detailed ablation study of $H_2$GCN (which is free of confounding designs) shows that the identified designs result in up to 40% performance gain in heterophily. In real networks, we observe that GNN models utilizing even a subset of our identified designs outperform popular models without them by up to 27% in heterophily, while being competitive in homophily. § 5

## 2 Notation and Preliminaries

We summarize our notation in Table A.1 (App. A). Let $\mathcal{G} = (\mathcal{V}, \mathcal{E})$ be an undirected, unweighted graph with nodeset $\mathcal{V}$ and edgeset $\mathcal{E}$. We denote a general neighborhood centered around $v$ as $N(v)$ ($\mathcal{G}$ may have self-loops), the corresponding neighborhood that does *not* include the ego (node $v$) as $\bar{N}(v)$, and the general neighbors of node $v$ at exactly $i$ hops/steps away (minimum distance) as $N_i(v)$. For example, $N_1(v) = \{u : (u, v) \in \mathcal{E}\}$ are the immediate neighbors of $v$. Other examples are shown in Fig. 1. We represent the graph by its adjacency matrix $\mathbf{A} \in \{0, 1\}^{n \times n}$ and its node feature matrix $\mathbf{X} \in \mathbb{R}^{n \times F}$, where the vector $\mathbf{x}_v$ corresponds to the *ego-feature* of node $v$, and $\{\mathbf{x}_u : u \in \bar{N}(v)\}$ to its *neighbor-features*.

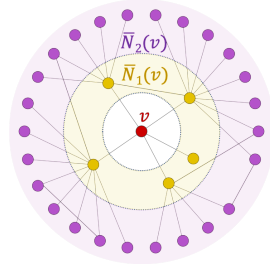

Figure 1: Neighborhoods.

We further assume a class label vector $\mathbf{y}$, which for each node $v$ contains a unique class label $y_v$. The goal of semi-supervised node classification is to learn a mapping $\ell : \mathcal{V} \to \mathcal{Y}$, where $\mathcal{Y}$ is the set of labels, given a set of labeled nodes $\mathcal{T}_{\mathcal{V}} = \{(v_1, y_1), (v_2, y_2), ...\}$ as training data.

**Graph neural networks**  From a probabilistic perspective, most GNN models assume the following local Markov property on node features: for each node $v \in \mathcal{V}$, there exists a neighborhood $N(v)$ such that $y_v$ only depends on the ego-feature $\mathbf{x}_v$ and neighbor-features $\{\mathbf{x}_u : u \in N(v)\}$. Most models derive the class label $y_v$ via the following representation learning approach:

$$\mathbf{r}_v^{(k)} = f\left(\mathbf{r}_v^{(k-1)}, \{\mathbf{r}_u^{(k-1)} : u \in N(v)\}\right), \ \mathbf{r}_v^{(0)} = \mathbf{x}_v, \text{ and } y_v = \arg\max\{\text{softmax}(\mathbf{r}_v^{(K)})\mathbf{W}\}, \quad (1)$$

where the embedding function $f$ is applied repeatedly in $K$ total rounds, node $v$'s representation (or hidden state vector) at round $k$, $\mathbf{r}_v^{(k)}$, is learned from its ego- and neighbor-representations in the previous round, and a softmax classifier with learnable weight matrix $\mathbf{W}$ is applied to the final representation of $v$. Most existing models differ in their definitions of neighborhoods $N(v)$ and embedding function $f$. A typical definition of neighborhood is $N_1(v)$—i.e., the 1-hop neighbors of $v$. As for $f$, in graph convolutional networks (GCN) [17] each node repeatedly averages its own features and those of its neighbors to update its own feature representation. Using an attention mechanism, GAT [36] models the influence of different neighbors more precisely as a weighted average of the ego- and neighbor-features. GraphSAGE [11] generalizes the aggregation beyond averaging, and models the ego-features distinctly from the neighbor-features in its subsampled neighborhood.

**Homophily and heterophily**  In this work, we focus on heterophily in class labels. We first define the edge homophily ratio $h$ as a measure of the graph homophily level, and use it to define graphs with strong homophily/heterophily:

**Definition 1** *The edge homophily ratio* $h = \frac{|\{(u,v):(u,v)\in\mathcal{E}\wedge y_u=y_v\}|}{|\mathcal{E}|}$ *is the fraction of edges in a graph which connect nodes that have the same class label (i.e., intra-class edges).*

**Definition 2** *Graphs with strong homophily have high edge homophily ratio* $h \to 1$*, while graphs with strong heterophily (i.e., low/weak homophily) have small edge homophily ratio* $h \to 0$*.*

The edge homophily ratio in Dfn. 1 gives an *overall trend* for all the edges in the graph. The actual level of homophily may vary within different pairs of node classes, i.e., there is different tendency of connection between each pair of classes. In App. B, we give more details about capturing these more complex network characteristics via an empirical *class compatibility matrix* **H**, whose $i, j$-th entry is the fraction of outgoing edges to nodes in class $j$ among all outgoing edges from nodes in class $i$.

*Heterophily $\neq$ Heterogeneity.* We remark that heterophily, which we study in this work, is a distinct network concept from heterogeneity. Formally, a network is heterogeneous [34] if it has at least two types of nodes and different relationships between them (e.g., knowledge graphs), and homogeneous if it has a single type of nodes (e.g., users) and a single type of edges (e.g., friendship). The type of nodes in heterogeneous graphs does *not* necessarily match the class labels $y_v$, therefore both homogeneous and heterogeneous networks may have different levels of homophily.

## 3 Learning Over Networks with Heterophily

While many GNN models have been proposed, most of them are designed under the assumption of homophily, and are not capable of handling heterophily. As a motivating example, Table 1 shows the mean classification accuracy for several leading GNN models on our synthetic benchmark `syn-cora`, where we can control the homophily/heterophily level (see App. G for details on the data and setup). Here we consider two homophily ratios, $h = 0.1$ and $h = 0.7$, one for high heterophily and one for high homophily. We observe that for heterophily ($h = 0.1$) all existing methods fail to perform better than a Multilayer Perceptron (MLP) with 1 hidden layer, a graph-agnostic baseline that relies solely on the node features for classification (differences in accuracy

Table 1: Example of a heterophily setting ($h = 0.1$) where existing GNNs fail to generalize, and a typical homophily setting ($h = 0.7$): mean accuracy and standard deviation over three runs (cf. App. G).

|  | $h = \mathbf{0.1}$ | $h = \mathbf{0.7}$ |
|---|---|---|
| GCN [17] | $37.14_{\pm 4.60}$ | $84.52_{\pm 0.54}$ |
| GAT [36] | $33.11_{\pm 1.20}$ | $84.03_{\pm 0.97}$ |
| GCN-Cheby [7] | $68.10_{\pm 1.75}$ | $84.92_{\pm 1.03}$ |
| GraphSAGE [11] | $72.89_{\pm 2.42}$ | $85.06_{\pm 0.51}$ |
| MixHop [1] | $58.93_{\pm 2.84}$ | $84.43_{\pm 0.94}$ |
| MLP | $74.85_{\pm 0.76}$ | $71.72_{\pm 0.62}$ |
| H$_2$GCN (**ours**) | $\mathbf{76.87_{\pm 0.43}}$ | $\mathbf{88.28_{\pm 0.66}}$ |

of MLP for different $h$ are due to randomness). Especially, GCN [17] and GAT [36] show up to 42% worse performance than MLP, highlighting that methods that work well under high homophily ($h = 0.7$) may not be appropriate for networks with low/medium homophily.

Motivated by this limitation, in the following subsections, we discuss and theoretically justify a set of key design choices that, when appropriately incorporated in a GNN framework, can improve the performance in the challenging heterophily settings. Then, we present H$_2$GCN, a model that, thanks to these designs, adapts well to both homophily and heterophily (Table 1, last row). In Section 5, we provide a comprehensive empirical analysis on both synthetic and real data with varying homophily levels, and show that the identified designs significantly improve the performance of GNNs (not limited to H$_2$GCN) by effectively leveraging the graph structure in challenging heterophily settings, while maintaining competitive performance in homophily.

### 3.1 Effective Designs for Networks with Heterophily

We have identified three key designs that—when appropriately integrated—can help improve the performance of GNN models in heterophily settings: (D1) ego- and neighbor-embedding separation; (D2) higher-order neighborhoods; and (D3) combination of intermediate representations. While these designs have been utilized separately in some prior works [11, 7, 1, 38], we are the first to discuss their importance *under heterophily* by providing novel theoretical justifications and an extensive empirical analysis on a variety of datasets.

#### 3.1.1 (D1) Ego- and Neighbor-embedding Separation

The first design entails encoding each ego-embedding (i.e., a node's embedding) *separately* from the aggregated embeddings of its neighbors, since they are likely to be dissimilar in heterophily settings. Formally, the representation (or hidden state vector) learned for each node $v$ at round $k$ is given as:

$$\mathbf{r}_v^{(k)} = \texttt{COMBINE}\left(\mathbf{r}_v^{(k-1)}, \texttt{AGGR}(\{\mathbf{r}_u^{(k-1)} : u \in \bar{N}(v)\})\right), \tag{2}$$

the neighborhood $\bar{N}(v)$ does *not* include $v$ (no self-loops), the `AGGR` function aggregates representations *only* from the neighbors (in some way—e.g., average), and `AGGR` and `COMBINE` may be followed

by a non-linear transformation. For heterophily, after aggregating the neighbors' representations, the definition of COMBINE (akin to 'skip connection' between layers) is critical: a simple way to combine the ego- and the aggregated neighbor-embeddings without 'mixing' them is with concatenation as in GraphSAGE [11]—rather than averaging *all* of them as in the GCN model by Kipf and Welling [17].

*Intuition.* In heterophily settings, by definition (Dfn. 2), the class label $y_v$ and original features $\mathbf{x}_v$ of a node and those of its neighboring nodes $\{(y_u, \mathbf{x}_u) : u \in \bar{N}(v)\}$ (esp. the direct neighbors $\bar{N}_1(v)$) may be different. However, the typical GCN design that mixes the embeddings through an average [17] or weighted average [36] as the COMBINE function results in final embeddings that are similar across neighboring nodes (especially within a community or cluster) *for any set of original features* [28]. While this may work well in the case of homophily, where neighbors likely belong to the same cluster and class, it poses severe challenges in the case of heterophily: it is not possible to distinguish neighbors from different classes based on the (similar) learned representations. Choosing a COMBINE function that separates the representations of each node $v$ and its neighbors $\bar{N}(v)$ allows for more expressiveness, where the skipped or non-aggregated representations can evolve separately over multiple rounds of propagation without becoming prohibitively similar.

*Theoretical Justification.* We prove theoretically that, under some conditions, a GCN layer that co-embeds ego- and neighbor-features is less capable of generalizing to heterophily than a layer that embeds them separately. We measure its generalization ability by its robustness to test/train data deviations. We give the proof of the theorem in App. C.1. Though the theorem applies to specific conditions, our empirical analysis shows that it holds in more general cases (§ 5).

**Theorem 1** *Consider a graph $\mathcal{G}$ without self-loops (§ 2) with node features $\mathbf{x}_v = \mathrm{onehot}(y_v)$ for each node $v$, and an equal number of nodes per class $y \in \mathcal{Y}$ in the training set $\mathcal{T}_{\mathcal{V}}$. Also assume that all nodes in $\mathcal{T}_{\mathcal{V}}$ have degree $d$, and proportion $h$ of their neighbors belong to the same class, while proportion $\frac{1-h}{|\mathcal{Y}|-1}$ of them belong to any other class (uniformly). Then for $h < \frac{1-|\mathcal{Y}|+2d}{2|\mathcal{Y}|d}$, a simple GCN layer formulated as $(\mathbf{A} + \mathbf{I})\mathbf{X}\mathbf{W}$ is less robust, i.e., misclassifies a node for smaller train/test data deviations, than a $\mathbf{A}\mathbf{X}\mathbf{W}$ layer that separates the ego- and neighbor-embeddings.*

*Observations.* In Table 1, we observe that GCN, GAT, and MixHop, which 'mix' the ego- and neighbor-embeddings explicitly[1], perform poorly in the heterophily setting. On the other hand, GraphSAGE that separates the embeddings (e.g., it concatenates the two embeddings and then applies a non-linear transformation) achieves 33-40% better performance in this setting.

### 3.1.2 (D2) Higher-order Neighborhoods

The second design involves explicitly aggregating information from higher-order neighborhoods in each round $k$, beyond the immediate neighbors of each node:

$$\mathbf{r}_v^{(k)} = \mathtt{COMBINE}\left(\mathbf{r}_v^{(k-1)}, \mathtt{AGGR}(\{\mathbf{r}_u^{(k-1)} : u \in \boxed{N_1(v)}\}), \mathtt{AGGR}(\{\mathbf{r}_u^{(k-1)} : u \in \boxed{N_2(v)}\}), \dots\right) \quad (3)$$

where $N_i(v)$ denotes the neighbors of $v$ at *exactly* $i$ hops away, and the AGGR functions applied to different neighborhoods can be the same or different. This design—employed in GCN-Cheby [7] and MixHop [1]—augments the *implicit* aggregation over higher-order neighborhoods that most GNN models achieve through multiple rounds of first-order propagation based on variants of Eq. (2).

*Intuition.* To show why higher-order neighborhoods help in the heterophily settings, we first define *homophily-dominant* and *heterophily-dominant* neighborhoods:

**Definition 3** $N(v)$ *is expectedly homophily-dominant if $P(y_u = y_v | y_v) \geq P(y_u = y | y_v), \forall u \in N(v)$ and $y \in \mathcal{Y} \neq y_v$. If the opposite inequality holds, $N(v)$ is expectedly heterophily-dominant.*

From this definition, we can see that expectedly homophily-dominant neighborhoods are more beneficial for GNN layers, as in such neighborhoods the class label $y_v$ of each node $v$ can *in expectation* be determined by the majority of the class labels in $N(v)$. In the case of heterophily, we have seen empirically that although the immediate neighborhoods may be heterophily-dominant, the higher-order neighborhoods may be homophily-dominant and thus provide more relevant context. This observation is also confirmed by recent works [2, 6] in the context of binary attribute prediction.

*Theoretical Justification.* Below we formalize the above observation for 2-hop neighborhoods under non-binary attributes (labels), and prove one case when they are homophily-dominant in App. C.2:

**Theorem 2** *Consider a graph $\mathcal{G}$ without self-loops (§ 2) with label set $\mathcal{Y}$, where for each node $v$, its neighbors' class labels $\{y_u : u \in N(v)\}$ are conditionally independent given $y_v$, and $P(y_u = y_v|y_v) = h$, $P(y_u = y|y_v) = \frac{1-h}{|\mathcal{Y}|-1}, \forall y \neq y_v$. Then, the 2-hop neighborhood $N_2(v)$ for a node $v$ will always be homophily-dominant in expectation.*

*Observations.* Under heterophily ($h = 0.1$), GCN-Cheby, which models different neighborhoods by combining Chebyshev polynomials to approximate a higher-order graph convolution operation [7], outperforms GCN and GAT, which aggregate over only the immediate neighbors $N_1$, by up to +31% (Table 1). MixHop, which explicitly models 1-hop and 2-hop neighborhoods (though 'mixes' the ego- and neighbor-embeddings[1], violating design D1), also outperforms these two models.

### 3.1.3 (D3) Combination of Intermediate Representations

The third design combines the intermediate representations of each node at the final layer:

$$\mathbf{r}_v^{(\text{final})} = \text{COMBINE}\left( \mathbf{r}_v^{(1)}, \mathbf{r}_v^{(2)}, \dots, \mathbf{r}_v^{(K)} \right) \tag{4}$$

to explicitly capture local *and* global information via COMBINE functions that leverage each representation separately–e.g., concatenation, LSTM-attention [38]. This design is introduced in jumping knowledge networks [38] and shown to increase the representation power of GCNs under *homophily*.

*Intuition.* Intuitively, each round collects information with different locality—earlier rounds are more local, while later rounds capture increasingly more global information (implicitly, via propagation). Similar to D2 (which models explicit neighborhoods), this design models the distribution of neighbor representations in low-homophily networks more accurately. It also allows the class prediction to leverage different neighborhood ranges in different networks, adapting to their structural properties.

*Theoretical Justification.* The benefit of combining intermediate representations can be theoretically explained from the spectral perspective. Assuming a GCN-style layer—where propagation can be viewed as spectral filtering—, the higher order polynomials of the normalized adjacency matrix $\mathbf{A}$ is a low-pass filter [37], so intermediate outputs from earlier rounds contain higher-frequency components than outputs from later rounds. At the same time, the following theorem holds for graphs with heterophily, where we view class labels as graph signals (as in graph signal processing):

**Theorem 3** *Consider graph signals (label vectors) $\mathbf{s}, \mathbf{t} \in \{0,1\}^{|\mathcal{V}|}$ defined on an undirected graph $\mathcal{G}$ with edge homophily ratios $h_s$ and $h_t$, respectively. If $h_s < h_t$, then signal $\mathbf{s}$ has higher energy (Dfn. 5) in high-frequency components than $\mathbf{t}$ in the spectrum of unnormalized graph Laplacian $\mathbf{L}$.*

In other words, in heterophily settings, the label distribution contains more information at higher than lower frequencies (see proof in App. C.3). Thus, by combining the intermediate outputs from different layers, this design captures both low- and high-frequency components in the final representation, which is critical in heterophily settings, and allows for more expressiveness in the general setting.

*Observations.* By concatenating the intermediate representations from two rounds with the embedded ego-representation (following the jumping knowledge framework [38]), GCN's accuracy increases to $58.93\%_{\pm 3.17}$ for $h = 0.1$, a 20% improvement over its counterpart without design D3 (Table 1).

**Summary of designs** To sum up, D1 models (at each layer) the ego- and neighbor-representations *distinctly*, D2 leverages (at each layer) representations of neighbors at different distances *distinctly*, and D3 leverages (at the final layer) the learned ego-representations at previous layers *distinctly*.

## 3.2 H$_2$GCN: A Framework for Networks with Homophily or Heterophily

We now describe H$_2$GCN, which exemplifies how effectively combining designs D1-D3 can help better adapt to the whole spectrum of low-to-high homophily, while avoiding interference with other designs. It has three stages (Alg. 1, App. D): **(S1)** feature embedding, **(S2)** neighborhood aggregation, and **(S3)** classification.

The *feature embedding stage* **(S1)** uses a graph-agnostic dense layer to generate for each node $v$ the feature embedding $\mathbf{r}_v^{(0)} \in \mathbb{R}^p$ based on its ego-feature $\mathbf{x}_v$: $\mathbf{r}_v^{(0)} = \sigma(\mathbf{x}_v \mathbf{W}_e)$, where $\sigma$ is an optional non-linear function, and $\mathbf{W}_e \in \mathbb{R}^{F \times p}$ is a learnable weight matrix.

In the *neighborhood aggregation stage* **(S2)**, the generated embeddings are aggregated and repeatedly updated within the node's neighborhood for $K$ rounds. Following designs D1 and D2, the neighborhood $N(v)$ of our framework involves two sub-neighborhoods without the egos: the 1-hop graph neighbors $\bar{N}_1(v)$ and the 2-hop neighbors $\bar{N}_2(v)$, as shown in Fig. 1:

$$\mathbf{r}_v^{(k)} = \texttt{COMBINE}\left(\texttt{AGGR}\{\mathbf{r}_u^{(k-1)} : u \in \bar{N}_1(v)\}, \texttt{AGGR}\{\mathbf{r}_u^{(k-1)} : u \in \bar{N}_2(v)\}\right). \quad (5)$$

We set `COMBINE` as concatenation (as to not mix different neighborhood ranges), and `AGGR` as a degree-normalized average of the neighbor-embeddings in sub-neighborhood $\bar{N}_i(v)$:

$$\mathbf{r}_v^{(k)} = \left(\mathbf{r}_{v,1}^{(k)} \| \mathbf{r}_{v,2}^{(k)}\right) \quad \text{and} \quad \mathbf{r}_{v,i}^{(k)} = \texttt{AGGR}\{\mathbf{r}_u^{(k-1)} : u \in \bar{N}_i(v)\} = \sum_{u \in \bar{N}_i(v)} \mathbf{r}_u^{(k-1)} d_{v,i}^{-1/2} d_{u,i}^{-1/2}, \quad (6)$$

where $d_{v,i} = |\bar{N}_i(v)|$ is the $i$-hop degree of node $v$ (i.e., number of nodes in its $i$-hop neighborhood). Unlike Eq. (2), here we do *not* combine the ego-embedding of node $v$ with the neighbor-embeddings. We found that removing the usual nonlinear transformations per round, as in SGC [37], works better (App. D.2), in which case we only need to include the ego-embedding in the final representation. By design D3, each node's final representation combines all its intermediate representations:

$$\mathbf{r}_v^{(\text{final})} = \texttt{COMBINE}\left(\mathbf{r}_v^{(0)}, \mathbf{r}_v^{(1)}, \ldots, \mathbf{r}_v^{(K)}\right), \quad (7)$$

where we empirically find concatenation works better than max-pooling [38] as the `COMBINE` function.

In the *classification stage* **(S3)**, the node is classified based on its final embedding $\mathbf{r}_v^{(\text{final})}$:

$$y_v = \arg\max\{\text{softmax}(\mathbf{r}_v^{(\text{final})} \mathbf{W}_c)\}, \quad (8)$$

where $\mathbf{W}_c \in \mathbb{R}^{(2^{K+1}-1)p \times |\mathcal{Y}|}$ is a learnable weight matrix. We visualize our framework in App. D.

**Time complexity**  The feature embedding stage **(S1)** takes $O(\text{nnz}(\mathbf{X})\,p)$, where $\text{nnz}(\mathbf{X})$ is the number of non-0s in feature matrix $\mathbf{X} \in \mathbb{R}^{n \times F}$, and $p$ is the dimension of the feature embeddings. The neighborhood aggregation stage **(S2)** takes $O(|\mathcal{E}|d_{\max})$ to derive the 2-hop neighborhoods via sparse-matrix multiplications, where $d_{\max}$ is the maximum degree of all nodes, and $O\left(2^K(|\mathcal{E}| + |\mathcal{E}_2|)p\right)$ for $K$ rounds of aggregation, where $|\mathcal{E}_2| = \frac{1}{2}\sum_{v \in \mathcal{V}} |\bar{N}_2(v)|$. We give a detailed analysis in App. D.

## 4  Other Related Work

We discuss relevant work on GNNs here, and give other related work (e.g., classification under heterophily) in Appendix E. Besides the models mentioned above, there are various comprehensive reviews describing previously proposed architectures [42, 5, 41]. Recent work has investigated GNN's ability to capture graph information, proposing diagnostic measurements based on feature smoothness and label smoothness [12] that may guide the learning process. To capture more graph information, other works generalize graph convolution outside of immediate neighborhoods. For example, apart from MixHop [1] (cf. § 3.1), Graph Diffusion Convolution [18] replaces the adjacency matrix with a sparsified version of a diffusion matrix (e.g., heat kernel or PageRank). Geom-GCN [26] precomputes unsupervised node embeddings and uses neighborhoods defined by geometric relationships in the resulting latent space to define graph convolution. Some of these works [1, 26, 12] acknowledge the challenges of learning from graphs with heterophily. Others have noted that node labels may have complex relationships that should be modeled directly. For instance, Graph Agreement Models [33] augment the classification task with an agreement task, co-training a model to predict whether pairs of nodes share the same label; Graph Markov Neural Networks [27] model the joint label distribution with a conditional random field, trained with expectation maximization using GNNs; Correlated Graph Neural Networks [15] model the correlation structure in the residuals of a regression task with a multivariate Gaussian, and can learn negative label correlations for neighbors in heterophily (for binary class labels); and the recent CPGNN [43] method models more complex label correlations by integrating the compatibility matrix notion from belief propagation [10] into GNNs.

**Comparison of H$_2$GCN to existing GNN models**  As shown in Table 2, H$_2$GCN differs from existing GNN models with respect to designs D1-D3, and their implementations (we give more details in App. D). Notably, H$_2$GCN learns a graph-agnostic feature embedding in stage **(S1)**, and skips the non-linear embeddings of aggregated representations per round that other models use (e.g., GraphSAGE, MixHop, GCN), resulting in a simpler yet powerful architecture.

Table 2: Design Comparison.

| Method | D1 | D2 | D3 |
|---|---|---|---|
| GCN [17] | ✗ | ✗ | ✗ |
| GAT [36] | ✗ | ✗ | ✗ |
| GCN-Cheby [7] | ✗ | ✓ | ✗ |
| GraphSAGE [11] | ✓ | ✗ | ✗ |
| MixHop [1] | ✗ | ✓ | ✗ |
| H$_2$GCN (proposed) | ✓ | ✓ | ✓ |

Table 3: Statistics for Synthetic Datasets

| Benchmark Name | #Nodes $|\mathcal{V}|$ | #Edges $|\mathcal{E}|$ | #Classes $|\mathcal{Y}|$ | #Features $F$ | Homophily $h$ | #Graphs |
|---|---|---|---|---|---|---|
| `syn-cora` | $1,490$ | $2,965$ to $2,968$ | $5$ | `cora` [30, 39] | $[0, 0.1, \ldots, 1]$ | $33$ (3 per $h$) |
| `syn-products` | $10,000$ | $59,640$ to $59,648$ | $10$ | `ogbn-products` [13] | $[0, 0.1, \ldots, 1]$ | $33$ (3 per $h$) |

# 5 Empirical Evaluation

We show the significance of designs D1-D3 on synthetic and real graphs with low-to-high homophily (Tab. 3, 5) via an ablation study of $H_2$GCN and comparison of models with and without the designs.

**Baseline models**  We consider MLP with 1 hidden layer, and all the methods listed in Table 2. For $H_2$GCN, we model the first- and second-order neighborhoods ($\bar{N}_1$ and $\bar{N}_2$), and consider two variants: $H_2$GCN-1 uses one embedding round ($K = 1$) and $H_2$GCN-2 uses two rounds ($K = 2$). We tune all the models on the same train/validation splits (see App. F for details).

## 5.1 Evaluation on Synthetic Benchmarks

**Synthetic datasets & setup**  We generate synthetic graphs with various homophily ratios $h$ (Tab. 3) by adopting an approach similar to [16]. In App. G, we describe the data generation process, the experimental setup, and the data statistics in detail. All methods share the same training, validation and test splits (25%, 25%, 50% per class), and we report the average accuracy and standard deviation (stdev) over three generated graphs per heterophily level and benchmark dataset.

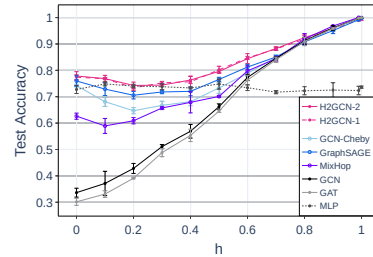

(a) `syn-cora` (Table G.2)

**Model comparison**  Figure 2 shows the mean test accuracy (and stdev) over all random splits of our synthetic benchmarks. We observe similar trends on both benchmarks: $H_2$GCN has the best trend overall, outperforming the baseline models in most heterophily settings, while tying with other models in homophily. The performance of GCN, GAT and MixHop, which mix the ego- and neighbor-embeddings, increases with respect to the homophily level. But, while they achieve near-perfect accuracy under strong homophily ($h \rightarrow 1$), they are significantly less accurate than MLP (near-flat performance curve as it is graph-agnostic) for many heterophily settings. GraphSAGE and GCN-Cheby, which leverage some of the identified designs D1-D3 (Table 2, § 3), are more competitive in such settings. We note that all the methods—except GCN and GAT—learn more effectively under perfect heterophily ($h$=0) than weaker settings (e.g., $h \in [0.1, 0.3]$), as evidenced by the J-shaped performance curves in low-homophily ranges.

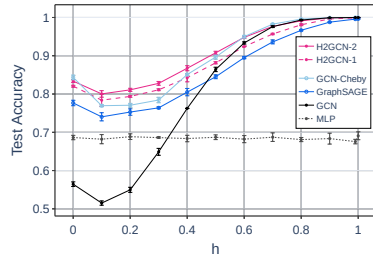

(b) `syn-products` (Table G.3). Mix-Hop acc $< 30\%$; GAT acc $< 50\%$ for $h < 0.4$.

Figure 2: Performance of GNN models on synthetic datasets. $H_2$GCN-2 outperforms baseline models in most heterophily settings, while tying with other models in homophily.

**Significance of design choices**  Using `syn-products`, we show the significance of designs D1-D3 (§ 3.1) through ablation studies with variants of $H_2$GCN (Fig. 3, Table G.4).

(D1) *Ego- and Neighbor-embedding Separation.*  We consider $H_2$GCN-1 variants that *separate* the ego- and neighbor-embeddings and model: (`S0`) neighborhoods $\bar{N}_1$ and $\bar{N}_2$ (i.e., $H_2$GCN-1); (`S1`) only the 1-hop neighborhood $\bar{N}_1$ in Eq. (5); and their counterparts that do *not separate* the two embeddings and use: (`NS0`) neighborhoods $N_1$ and $N_2$ (including $v$); and (`NS1`) only the 1-hop neighborhood $N_1$. Figure 3a shows that the variants that learn separate embedding functions significantly outperform the others (`NS0/1`) in heterophily settings ($h < 0.7$) by up to $40\%$, which shows that design D1 is critical for success in heterophily. $H_2$GCN-1 (`S0`) performs best in homophily.

(D2) *Higher-order Neighborhoods.*  For this design, we consider three variants of $H_2$GCN-1 without specific neighborhoods: (`N0`) without the 0-hop neighborhood $N_0(v) = v$ (i.e, the ego-embedding) (`N1`) without $\bar{N}_1(v)$; and (`N2`) without $\bar{N}_2(v)$. Figure 3b shows that $H_2$GCN-1 consistently performs better than all the variants, indicating that combining all sub-neighborhoods works best. Among the variants, in heterophily settings, $N_0(v)$ contributes most to the performance (`N0` causes significant decrease in accuracy), followed by $\bar{N}_1(v)$, and $\bar{N}_2(v)$. However, when $h \geq 0.7$, the importance of sub-neighborhoods is reversed. Thus, the ego-features are the most important in heterophily, and

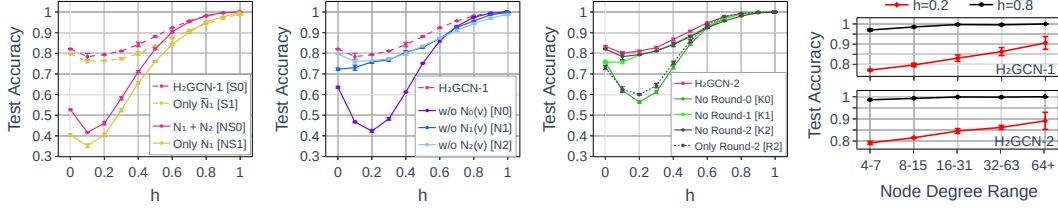

(a) Design D1: Embed-   (b) Design D2: Higher-   (c) Design D3: Intermedi-   (d) Accuracy per degree in
ding separation.            order neighborhoods.         ate representations.           hetero/homo-phily.

Figure 3: (a)-(c): Significance of design choices D1-D3 via ablation studies. (d): Performance of
$H_2$GCN for different node degree ranges. In heterophily, the performance gap between low- and
high-degree nodes is significantly larger than in homophily, i.e., low-degree nodes pose challenges.

higher-order neighborhoods contribute the most in homophily. The design of $H_2$GCN allows it to
effectively combine information from different neighborhoods, adapting to all levels of homophily.

(D3) *Combination of Intermediate Representations.* We consider three variants (K-0,1,2) of $H_2$GCN-2
that drop from the final representation of Eq. (7) the $0^{th}$, $1^{st}$ or $2^{nd}$-round intermediate representation,
respectively. We also consider only the $2^{nd}$ intermediate representation as final, which is akin to what
the other GNN models do. Figure 3c shows that $H_2$GCN-2, which combines all the intermediate
representations, performs the best, followed by the variant K2 that skips the round-2 representation.
The ego-embedding is the most important for heterophily $h \leq 0.5$ (see trend of K0).

**The challenging case of low-degree nodes**  Figure 3d plots the mean accuracy of $H_2$GCN variants
on `syn-products` for different node degree ranges both in a heterophily and a homophily setting
($h \in \{0.2, 0.8\}$). We observe that under heterophily there is a significantly bigger performance gap
between low- and high-degree nodes: 13% for $H_2$GCN-1 (10% for $H_2$GCN-2) vs. less than 3%
under homophily. This is likely due to the importance of the *distribution* of class labels in each
neighborhood under heterophily, which is harder to estimate accurately for low-degree nodes with
few neighbors. On the other hand, in homophily, neighbors are likely to have similar classes $y \in \mathcal{Y}$,
so the neighborhood size does not have as significant impact on the accuracy.

## 5.2   Evaluation on Real Benchmarks

**Real datasets & setup**   We now evaluate the performance of our model and existing GNNs on a variety
of real-world datasets [35, 29, 30, 22, 4, 31] with edge
homophily ratio $h$ ranging from strong heterophily
to strong homophily, going beyond the traditional
Cora, Pubmed and Citeseer graphs that have strong
homophily (hence the good performance of existing
GNNs on them). We summarize the data in Table 5,
and describe them in App. H, where we also point
out potential data limitations. For all benchmarks (ex-
cept `Cora-Full`), we use the feature vectors, class
labels, and 10 random splits (48%/32%/20% of nodes
per class for train/validation/test[2]) provided by [26].
For Cora-Full, we generate 3 random splits, with
25%/25%/50% of nodes per class for train/valida-
tion/test.

Table 4: Real benchmarks: Average rank per
method (and their employed designs among
D1-D3) under heterophily (benchmarks with
$h \leq 0.3$), homophily ($h \geq 0.7$), and across
the full spectrum ("Overall"). The "*" de-
notes ranks based on results reported in [26].

| Method (Designs) | Het. | Hom. | Overall |
|---|---|---|---|
| **$H_2$GCN-1** (D1, D2, D3) | 3.8 | 3.0 | 3.6 |
| **$H_2$GCN-2** (D1, D2, D3) | 4.0 | 2.0 | 3.3 |
| **GraphSAGE** (D1) | 5.0 | 6.0 | 5.3 |
| **GCN-Cheby** (D2) | 7.0 | 6.3 | 6.8 |
| **MixHop** (D2) | 6.5 | 6.0 | 6.3 |
| **GraphSAGE+JK** (D1, D3) | 5.0 | 7.0 | 5.7 |
| **GCN-Cheby+JK** (D2, D3) | 3.7 | 7.7 | 5.0 |
| **GCN+JK** (D3) | 7.2 | 8.7 | 7.7 |
| **GCN** | 9.8 | 5.3 | 8.3 |
| **GAT** | 11.5 | 10.7 | 11.2 |
| **GEOM-GCN*** | 8.2 | 4.0 | 6.8 |
| **MLP** | 6.2 | 11.3 | 7.9 |

**Effectiveness of design choices**   Table 4 gives the
average ranks of our $H_2$GCN variants and other models on real benchmarks with heterophily,
homophily, and across the full spectrum. Table 5 gives detailed results (mean accuracy and stdev)
per benchmark. We observe that models which utilize all or subsets of our identified designs D1-D3
(§ 3.1) perform significantly better than GCN and GAT which lack these designs, especially in
heterophily. Next, we discuss the effectiveness of each design.

(D1) *Ego- and Neighbor-embedding Separation.* We compare GraphSAGE, which *separates* the
ego- and neighbor-embeddings, and GCN that does not. In heterophily settings, GraphSAGE has

Table 5: Real data: mean accuracy ± stdev over different data splits. Best model per benchmark highlighted in gray. The "*" results are obtained from [26] and "N/A" denotes non-reported results.

| | Texas | Wisconsin | Actor | Squirrel | Chameleon | Cornell | Cora Full | Citeseer | Pubmed | Cora |
|---|---|---|---|---|---|---|---|---|---|---|
| Hom. ratio $h$ | 0.11 | 0.21 | 0.22 | 0.22 | 0.23 | 0.3 | 0.57 | 0.74 | 0.8 | 0.81 |
| #Nodes $|\mathcal{V}|$ | 183 | 251 | 7,600 | 5,201 | 2,277 | 183 | 19,793 | 3,327 | 19,717 | 2,708 |
| #Edges $|\mathcal{E}|$ | 295 | 466 | 26,752 | 198,493 | 31,421 | 280 | 63,421 | 4,676 | 44,327 | 5,278 |
| #Classes $|\mathcal{Y}|$ | 5 | 5 | 5 | 5 | 5 | 5 | 70 | 7 | 3 | 6 |
| $H_2$GCN-1 | 84.86±6.77 | 86.67±4.69 | 35.86±1.03 | 36.42±1.89 | 57.11±1.58 | 82.16±4.80 | 68.13±0.49 | 77.07±1.64 | 89.40±0.34 | 86.92±1.37 |
| $H_2$GCN-2 | 82.16±5.28 | 85.88±4.22 | 35.62±1.30 | 37.90±2.02 | 59.39±1.98 | 82.16±6.00 | 69.05±0.37 | 76.88±1.77 | 89.59±0.33 | 87.81±1.35 |
| GraphSAGE | 82.43±6.14 | 81.18±5.56 | 34.23±0.99 | 41.61±0.74 | 58.73±1.68 | 75.95±5.01 | 65.14±0.75 | 76.04±1.30 | 88.45±0.50 | 86.90±1.04 |
| GCN-Cheby | 77.30±4.07 | 79.41±4.46 | 34.11±1.09 | 43.86±1.64 | 55.24±2.76 | 74.32±7.46 | 67.41±0.69 | 75.82±1.53 | 88.72±0.55 | 86.76±0.95 |
| MixHop | 77.84±7.73 | 75.88±4.90 | 32.22±2.34 | 43.80±1.48 | 60.50±2.53 | 73.51±6.34 | 65.59±0.34 | 76.26±1.33 | 85.31±0.61 | 87.61±0.85 |
| GraphSAGE+JK | 83.78±2.21 | 81.96±4.96 | 34.28±1.01 | 40.85±1.29 | 58.11±1.97 | 75.68±4.03 | 65.31±0.58 | 76.05±1.37 | 88.34±0.62 | 85.96±0.83 |
| Cheby+JK | 78.38±6.37 | 82.55±4.57 | 35.14±1.37 | 45.03±1.73 | 63.79±2.27 | 74.59±7.87 | 66.87±0.29 | 74.98±1.18 | 89.07±0.30 | 85.49±1.27 |
| GCN+JK | 66.49±6.64 | 74.31±6.43 | 34.18±0.85 | 40.45±1.61 | 63.42±2.00 | 64.59±8.68 | 66.72±0.61 | 74.51±1.75 | 88.41±0.45 | 85.79±0.92 |
| GCN | 59.46±5.25 | 59.80±6.99 | 30.26±0.79 | 36.89±1.34 | 59.82±2.58 | 57.03±4.67 | 68.39±0.32 | 76.68±1.64 | 87.38±0.66 | 87.28±1.26 |
| GAT | 58.38±4.45 | 55.29±8.71 | 26.28±1.73 | 30.62±2.11 | 54.69±1.95 | 58.92±3.32 | 59.81±0.92 | 75.46±1.72 | 84.68±0.44 | 82.68±1.80 |
| GEOM-GCN* | 67.57 | 64.12 | 31.63 | 38.14 | 60.90 | 60.81 | N/A | 77.99 | 90.05 | 85.27 |
| MLP | 81.89±4.78 | 85.29±3.61 | 35.76±0.98 | 29.68±1.81 | 46.36±2.52 | 81.08±6.37 | 58.76±0.50 | 72.41±2.18 | 86.65±0.35 | 74.75±2.22 |

an average rank of 5.0 compared to 9.8 for GCN, and outperforms GCN in almost all heterophily benchmarks by up to 23%. In homophily settings ($h \geq 0.7$), GraphSAGE ranks close to GCN (6.0 vs. 5.3), and GCN never outperforms GraphSAGE by more than 1% in mean accuracy. These results support the importance of D1 for success in heterophily and comparable performance in homophily.

(D2) *Higher-order Neighborhoods.* To show the benefits of design D2 under heterophily, we compare the performance of GCN-Cheby and MixHop—which define higher-order graph convolutions—to that of (first-order) GCN. Under heterophily, GCN-Cheby (rank 7.0) and MixHop (rank 6.5) have better performance than GCN (rank 9.8), and outperform the latter in all but one heterophily benchmarks by up to 20%. In most homophily benchmarks, the performance difference between these methods is less than 1%. Our observations highlight the importance of D2, especially in heterophily.

(D3) *Combination of Intermediate Representations.* We compare GraphSAGE, GCN-Cheby and GCN to their corresponding variants enhanced with JK connections [38]. GCN and GCN-Cheby benefit significantly from D3 in heterophily: their average ranks improve (9.8 vs. 7.2 and 7 vs 3.7, respectively) and their mean accuracies increase by up to 14% and 8%, respectively, in heterophily benchmarks. Though GraphSAGE+JK performs better than GraphSAGE on half of the heterophily benchmarks, its average rank remains unchanged. This may be due to the marginal benefit of D3 when combined with D1, which GraphSAGE employs. Under homophily, the performance with and without JK connections is similar (gaps mostly less than 2%), matching the observations in [38].

While other design choices and implementation details may confound a comparative evaluation of D1-D3 in different models (motivating our introduction of $H_2$GCN and our ablation study in § 3.1), these observations support the effectiveness of our identified designs on diverse GNN architectures and real-world datasets, and affirm our findings in the ablation study. We also observe that our $H_2$GCN variants, which combine the three identified designs, have consistently strong performance across the *full spectrum* of low-to-high homophily: $H_2$GCN-2 achieves the best average rank (3.3) across all datasets (or homophily ratios $h$), followed by $H_2$GCN-1 (3.6).

**Additional model comparison** In Table 4, we also report the *best* results among the *three* recently-proposed GEOM-GCN variants (§ 4), directly from the paper [26]: other models (including ours) outperform this method significantly under heterophily. We note that MLP is a competitive baseline under heterophily (ranked 6.2), indicating that many existing models do not use the graph information effectively, or the latter is misleading in such cases. All models perform poorly on Squirrel and Actor likely due to their low-quality node features (small correlation with class labels). Also, Squirrel and Chameleon are dense, with many nodes sharing the same neighbors.

## 6 Conclusion

We have focused on characterizing the representation power of GNNs in challenging settings with heterophily or low homophily, which is understudied in the literature. We have highlighted the current limitations of GNNs, presented designs that increase representation power under heterophily and are theoretically justified with perturbation analysis and graph signal processing, and introduced the $H_2$GCN model that adapts to both heterophily and homophily by effectively synthetizing these designs. We analyzed various challenging datasets, going beyond the often-used benchmark datasets (Cora, Pubmed, Citeseer), and leave as future work extending to a larger-scale experimental testbed.

## Broader Impact

Homophily and heterophily are not intrinsically ethical or unethical—they are both phenomena existing in the nature, resulting in the popular proverbs "birds of a feather flock together" and "opposites attract". However, many popular GNN models implicitly assume homophily; as a result, if they are applied to networks that do not satisfy the assumption, the results may be biased, unfair, or erroneous. In some applications, the homophily assumption may have ethical implications. For example, a GNN model that intrinsically assumes homophily may contribute to the so-called "filter bubble" phenomenon in a recommendation system (reinforcing existing beliefs/views, and downplaying the opposite ones), or make minority groups less visible in social networks. In other cases, a reliance on homophily may hinder scientific progress. Among other domains, this is critical for applying GNN models to molecular and protein structures, where the connected nodes often belong to different classes, and thus successful methods will need to model heterophily successfully.

Our work has the potential to rectify some of these potential negative consequences of existing GNN work. While our methodology does not change the amount of homophily in a network, moving beyond a reliance on homophily can be a key to improve the fairness, diversity and performance in applications using GNNs. We hope that this paper will raise more awareness and discussions regarding the homophily limitations of existing GNN models, and help researchers design models which have the power of learning in both homophily and heterophily settings.

## Acknowledgments and Disclosure of Funding

We thank the reviewers for their constructive feedback. This material is based upon work supported by the National Science Foundation under CAREER Grant No. IIS 1845491 and 1452425, Army Young Investigator Award No. W911NF1810397, an Adobe Digital Experience research faculty award, an Amazon faculty award, a Google faculty award, and AWS Cloud Credits for Research. We gratefully acknowledge the support of NVIDIA Corporation with the donation of the Quadro P6000 GPU used for this research. Any opinions, findings, and conclusions or recommendations expressed in this material are those of the author(s) and do not necessarily reflect the views of the National Science Foundation or other funding parties.

## Footnotes

[1] These models consider self-loops, which turn each ego also into a neighbor, and thus mix the ego- and neighbor-representations. E.g., GCN and MixHop operate on the symmetric normalized adjacency matrix augmented with self-loops: $\hat{\mathbf{A}} = \hat{\mathbf{D}}^{-\frac{1}{2}}(\mathbf{A} + \mathbf{I})\hat{\mathbf{D}}^{-\frac{1}{2}}$, where $\mathbf{I}$ is the identity and $\hat{\mathbf{D}}$ the degree matrix of $\mathbf{A} + \mathbf{I}$.

[2][26] claims that the ratios are 60%/20%/20%, which is different from the actual data splits shared on GitHub.

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
