[Supplementary Material]

# A Nomenclature

We summarize the main symbols used in this work and their definitions below:

Table A.1: Major symbols and definitions.

| Symbols | Definitions |
|---|---|
| $\mathcal{G} = (\mathcal{V}, \mathcal{E})$ | graph $\mathcal{G}$ with nodeset $\mathcal{V}$, edgeset $\mathcal{E}$ |
| $\mathbf{A}$ | $n \times n$ adjacency matrix of $\mathcal{G}$ |
| $\mathbf{X}$ | $n \times F$ node feature matrix of $\mathcal{G}$ |
| $\mathbf{x}_v$ | $F$-dimensional feature vector for node $v$ |
| $\mathbf{L}$ | unnormalized graph Laplacian matrix |
| $\mathcal{Y}$ | set of class labels |
| $y_v$ | class label for node $v \in \mathcal{V}$ |
| $\mathbf{y}$ | $n$-dimensional vector of class labels (for all the nodes) |
| $\mathcal{T}_\mathcal{V} = \{(v_1, y_1), (v_2, y_2), ...\}$ | training data for semi-supervised node classification |
| $N(v)$ | general type of neighbors of node $v$ in graph $\mathcal{G}$ |
| $\bar{N}(v)$ | general type of neighbors of node $v$ in $\mathcal{G}$ *without self-loops* (i.e., excluding $v$) |
| $N_i(v), \bar{N}_i(v)$ | $i$-hop/step neighbors of node $v$ in $\mathcal{G}$ (at exactly distance $i$) maybe-with/without self-loops, resp. |
| $\mathcal{E}_2$ | set of pairs of nodes $(u, v)$ with shortest distance between them being 2 |
| $d, d_{\max}$ | node degree, and maximum node degree across all nodes $v \in \mathcal{V}$, resp. |
| $h$ | edge homophily ratio |
| $\mathbf{H}$ | class compatibility matrix |
| $\mathbf{r}_v^{(k)}$ | node representations learned in GNN model at round / layer $k$ |
| $K$ | the number of rounds in the neighborhood aggregation stage |
| $\mathbf{W}$ | learnable weight matrix for GNN model |
| $\sigma$ | non-linear activation function |
| $\|$ | vector concatenation operator |
| AGGR | function that aggregates node feature representations within a neighborhood |
| COMBINE | function that combines feature representations from different neighborhoods |

# B Homophily and Heterophily: Compatibility Matrix

As we mentioned in § 2, the edge homophily ratio in Definition 1 gives an *overall trend* for all the edges in the graph. The actual level of homophily may vary within different pairs of node classes, i.e., there is different tendency of connection between each pair of classes. For instance, in an online purchasing network [24] with three classes—fraudsters, accomplices, and honest users—, fraudsters connect with higher probability to accomplices and honest users. Moreover, within the same network, it is possible that some pairs of classes exhibit homophily, while others exhibit heterophily. In belief propagation [40], a message-passing algorithm used for inference on graphical models, the different levels of homophily or affinity between classes are captured via the *class compatibility*, *propagation* or *coupling* matrix, which is typically pre-defined based on domain knowledge. In this work, we define the empirical *class compatibility matrix* $\mathbf{H}$ as follows:

**Definition 4** *The class compatibility matrix* $\mathbf{H}$ *has entries* $[\mathbf{H}]_{i,j}$ *that capture the fraction of outgoing edges from a node in class $i$ to a node in class $j$:*

$$[\mathbf{H}]_{i,j} = \frac{|\{(u, v) : (u, v) \in \mathcal{E} \land y_u = i \land y_v = j\}|}{|\{(u, v) : (u, v) \in \mathcal{E} \land y_u = i\}|}$$

By definition, the class compatibility matrix is a stochastic matrix, with each row summing up to 1.

# C  Proofs and Discussions of Theorems

## C.1  Detailed Analysis of Theorem 1

**Proof 1 (for Theorem 1)** *We first discuss the GCN layer formulated as $f(\mathbf{X}; \mathbf{A}, \mathbf{W}) = (\mathbf{A}+\mathbf{I})\mathbf{X}\mathbf{W}$. Given training set $\mathcal{T_V}$, the goal of the training process is to optimize the weight matrix $\mathbf{W}$ to minimize the loss function $\mathcal{L}([(\mathbf{A}+\mathbf{I})\mathbf{X}]_{\mathcal{T_V},:}\mathbf{W}, [\mathbf{Y}]_{\mathcal{T_V},:})$, where $[\mathbf{Y}]_{\mathcal{T_V},:}$ is the one-hot encoding of class labels provided in the training set, and $[(\mathbf{A}+\mathbf{I})\mathbf{X}]_{\mathcal{T_V},:}\mathbf{W}$ is the predicted probability distribution of class labels for each node $v$ in the training set $\mathcal{T_V}$.*

*Without loss of generality, we reorder $\mathcal{T_V}$ accordingly such that the one-hot encoding of labels for nodes in training set $[\mathbf{Y}]_{\mathcal{T_V},:}$ is in increasing order of the class label $y_v$:*

$$[\mathbf{Y}]_{\mathcal{T_V},:} = \begin{bmatrix} 1 & 0 & 0 & \cdots & 0 \\ \vdots & \vdots & \vdots & \ddots & \vdots \\ 1 & 0 & 0 & \cdots & 0 \\ \hdashline 0 & 1 & 0 & \cdots & 0 \\ \vdots & \vdots & \vdots & \ddots & \vdots \\ 0 & 1 & 0 & \cdots & 0 \\ \hdashline \vdots & \vdots & \vdots & \ddots & \vdots \\ \hdashline 0 & 0 & 0 & \cdots & 1 \\ \vdots & \vdots & \vdots & \ddots & \vdots \\ 0 & 0 & 0 & \cdots & 1 \end{bmatrix}_{|\mathcal{V}| \times |\mathcal{Y}|} \tag{9}$$

*Now we look into the term $[(\mathbf{A} + \mathbf{I})\mathbf{X}]_{\mathcal{T_V},:}$, which is the aggregated feature vectors within neighborhood $N_1$ for nodes in the training set. Since we assumed that all nodes in $\mathcal{T_V}$ have degree $d$, proportion $h$ of their neighbors belong to the same class, while proportion $\frac{1-h}{|\mathcal{Y}|-1}$ of them belong to any other class uniformly, and one-hot representations of node features $\mathbf{x}_v = \text{onehot}(y_v)$ for each node $v$, we obtain:*

$$[(\mathbf{A} + \mathbf{I})\mathbf{X}]_{\mathcal{T_V},:} = \begin{bmatrix} hd+1 & \frac{1-h}{|\mathcal{Y}|-1}d & \frac{1-h}{|\mathcal{Y}|-1}d & \cdots & \frac{1-h}{|\mathcal{Y}|-1}d \\ \vdots & \vdots & \vdots & \ddots & \vdots \\ hd+1 & \frac{1-h}{|\mathcal{Y}|-1}d & \frac{1-h}{|\mathcal{Y}|-1}d & \cdots & \frac{1-h}{|\mathcal{Y}|-1}d \\ \hdashline \frac{1-h}{|\mathcal{Y}|-1}d & hd+1 & \frac{1-h}{|\mathcal{Y}|-1}d & \cdots & \frac{1-h}{|\mathcal{Y}|-1}d \\ \vdots & \vdots & \vdots & \ddots & \vdots \\ \frac{1-h}{|\mathcal{Y}|-1}d & hd+1 & \frac{1-h}{|\mathcal{Y}|-1}d & \cdots & \frac{1-h}{|\mathcal{Y}|-1}d \\ \hdashline \vdots & \vdots & \vdots & \ddots & \vdots \\ \hdashline \frac{1-h}{|\mathcal{Y}|-1}d & \frac{1-h}{|\mathcal{Y}|-1}d & \frac{1-h}{|\mathcal{Y}|-1}d & \cdots & hd+1 \\ \vdots & \vdots & \vdots & \ddots & \vdots \\ \frac{1-h}{|\mathcal{Y}|-1}d & \frac{1-h}{|\mathcal{Y}|-1}d & \frac{1-h}{|\mathcal{Y}|-1}d & \cdots & hd+1 \end{bmatrix}_{|\mathcal{V}| \times |\mathcal{Y}|} \tag{10}$$

*For $[\mathbf{Y}]_{\mathcal{T_V},:}$ and $[(\mathbf{A}+\mathbf{I})\mathbf{X}]_{\mathcal{T_V},:}$ that we derived in Eq. (9) and (10), we can find an optimal weight matrix $\mathbf{W}_*$ such that $[(\mathbf{A}+\mathbf{I})\mathbf{X}]_{\mathcal{T_V},:}\mathbf{W}_* = [\mathbf{Y}]_{\mathcal{T_V},:}$, making the loss $\mathcal{L}([(\mathbf{A}+\mathbf{I})\mathbf{X}]_{\mathcal{T_V},:}\mathbf{W}_*, [\mathbf{Y}]_{\mathcal{T_V},:}) = 0$. We can use the following way to find $\mathbf{W}_*$: First, sample one node from each class to form a smaller*

*set $\mathcal{T}_S \subset \mathcal{T}_\mathcal{V}$, therefore we have:*

$$[\mathbf{Y}]_{\mathcal{T}_S,:} = \begin{bmatrix} 1 & 0 & 0 & \cdots & 0 \\ 0 & 1 & 0 & \cdots & 0 \\ \vdots & \vdots & \vdots & \ddots & \vdots \\ 0 & 0 & 0 & \cdots & 1 \end{bmatrix}_{|\mathcal{Y}|\times|\mathcal{Y}|} = \mathbf{I}_{|\mathcal{Y}|\times|\mathcal{Y}|}$$

*and*

$$[(\mathbf{A}+\mathbf{I})\mathbf{X}]_{\mathcal{T}_S,:} = \begin{bmatrix} hd+1 & \frac{1-h}{|\mathcal{Y}|-1}d & \frac{1-h}{|\mathcal{Y}|-1}d & \cdots & \frac{1-h}{|\mathcal{Y}|-1}d \\ \frac{1-h}{|\mathcal{Y}|-1}d & hd+1 & \frac{1-h}{|\mathcal{Y}|-1}d & \cdots & \frac{1-h}{|\mathcal{Y}|-1}d \\ \vdots & \vdots & \vdots & \ddots & \vdots \\ \frac{1-h}{|\mathcal{Y}|-1}d & \frac{1-h}{|\mathcal{Y}|-1}d & \frac{1-h}{|\mathcal{Y}|-1}d & \cdots & hd+1 \end{bmatrix}_{|\mathcal{Y}|\times|\mathcal{Y}|}$$

*Note that $[(\mathbf{A}+\mathbf{I})\mathbf{X}]_{\mathcal{T}_S,:}$ is a circulant matrix, therefore its inverse exists. Using the Sherman-Morrison formula, we can find its inverse as:*

$$([(\mathbf{A}+\mathbf{I})\mathbf{X}]_{\mathcal{T}_S,:})^{-1} = \frac{1}{(d+1)(|\mathcal{Y}|-1+(|\mathcal{Y}|h-1)d)} \cdot$$

$$\begin{bmatrix} (|\mathcal{Y}|-1)+(|\mathcal{Y}|-2+h)d & (h-1)d & \cdots & (h-1)d \\ (h-1)d & (|\mathcal{Y}|-1)+(|\mathcal{Y}|-2+h)d & \cdots & (h-1)d \\ \vdots & \vdots & \ddots & \vdots \\ (h-1)d & (h-1)d & \cdots & (|\mathcal{Y}|-1)+(|\mathcal{Y}|-2+h)d \end{bmatrix}$$

*Let $\mathbf{W}_* = ([(\mathbf{A}+\mathbf{I})\mathbf{X}]_{\mathcal{T}_S,:})^{-1}$, and we have $[(\mathbf{A}+\mathbf{I})\mathbf{X}]_{\mathcal{T}_S,:}\mathbf{W}_* = [\mathbf{Y}]_{\mathcal{T}_S,:} = \mathbf{I}_{|\mathcal{Y}|\times|\mathcal{Y}|}$. It is also easy to verify that $[(\mathbf{A}+\mathbf{I})\mathbf{X}]_{\mathcal{T}_\mathcal{V},:}\mathbf{W}_* = [\mathbf{Y}]_{\mathcal{T}_\mathcal{V},:}$. $\mathbf{W}_* = ([(\mathbf{A}+\mathbf{I})\mathbf{X}]_{\mathcal{T}_S,:})^{-1}$ is the optimal weight matrix we can learn under $\mathcal{T}_\mathcal{V}$, since it satisfies $\mathcal{L}([(\mathbf{A}+\mathbf{I})\mathbf{X}]_{\mathcal{T}_\mathcal{V},:}\mathbf{W}_*, [\mathbf{Y}]_{\mathcal{T}_\mathcal{V},:}) = 0$.*

*Now consider an arbitrary training datapoint $(v, y_v) \in \mathcal{T}_\mathcal{V}$, and a perturbation added to the neighborhood $N(v)$ of node $v$, such that the number of nodes with a randomly selected class label $y_p \in \mathcal{Y} \neq y_v$ is $\delta_1$ less than expected in $N(v)$. We denote the perturbed graph adjacency matrix as $\mathbf{A}_\triangle$. Without loss of generality, we assume node $v$ has $y_v = 1$, and the perturbed class is $y_p = 2$. In this case we have*

$$[(\mathbf{A}_\triangle+\mathbf{I})\mathbf{X}]_{v,:} = \begin{bmatrix} hd+1 & \frac{1-h}{|\mathcal{Y}|-1}d - \delta_1 & \frac{1-h}{|\mathcal{Y}|-1}d & \cdots & \frac{1-h}{|\mathcal{Y}|-1}d \end{bmatrix}$$

*Applying the optimal weight matrix we learned on $\mathcal{T}_\mathcal{V}$ to the aggregated feature on the perturbed neighborhood $[(\mathbf{A}_\triangle+\mathbf{I})\mathbf{X}]_{v,:}$, we obtain $[(\mathbf{A}_\triangle+\mathbf{I})\mathbf{X}]_{v,:}\mathbf{W}_*$ which equals to:*

$$\begin{bmatrix} 1 - \frac{(h-1)d\delta_1}{(d+1)(|\mathcal{Y}|-1+(|\mathcal{Y}|h-1)d)} & -\frac{((|\mathcal{Y}|-1)+(|\mathcal{Y}|-2+h)d)\delta_1}{(d+1)(|\mathcal{Y}|-1+(|\mathcal{Y}|h-1)d)} & -\frac{(h-1)d\delta_1}{(d+1)(|\mathcal{Y}|-1+(|\mathcal{Y}|h-1)d)} & \cdots & -\frac{(h-1)d\delta_1}{(d+1)(|\mathcal{Y}|-1+(|\mathcal{Y}|h-1)d)} \end{bmatrix}$$

*Notice that we always have $1 - \frac{(h-1)d\delta_1}{(d+1)(|\mathcal{Y}|-1+(|\mathcal{Y}|h-1)d)} > -\frac{(h-1)d\delta_1}{(d+1)(|\mathcal{Y}|-1+(|\mathcal{Y}|h-1)d)}$, thus the GCN layer formulated as $(\mathbf{A}+\mathbf{I})\mathbf{X}\mathbf{W}$ would misclassify only if the following inequality holds:*

$$1 - \frac{(h-1)d\delta_1}{(d+1)(|\mathcal{Y}|-1+(|\mathcal{Y}|h-1)d)} < -\frac{((|\mathcal{Y}|-1)+(|\mathcal{Y}|-2+h)d)\delta_1}{(d+1)(|\mathcal{Y}|-1+(|\mathcal{Y}|h-1)d)}$$

*Solving the above inequality for $\delta_1$, we get the amount of perturbation needed as*

$$\begin{cases} \delta_1 > \frac{-h|\mathcal{Y}|d-|\mathcal{Y}|+d+1}{|\mathcal{Y}|-1}, \text{ when } 0 \leq h < \frac{-|\mathcal{Y}|+d+1}{|\mathcal{Y}|d} \\ \delta_1 < \frac{-h|\mathcal{Y}|d-|\mathcal{Y}|+d+1}{|\mathcal{Y}|-1}, \text{ when } h > \frac{-|\mathcal{Y}|+d+1}{|\mathcal{Y}|d} \end{cases} \tag{11}$$

*and the least absolute amount of perturbation needed is $|\delta_1| = |\frac{-h|\mathcal{Y}|d-|\mathcal{Y}|+d+1}{|\mathcal{Y}|-1}|$.*

*Now we move on to discuss the GCN layer formulated as $f(\mathbf{X}; \mathbf{A}, \mathbf{W}) = \mathbf{A}\mathbf{X}\mathbf{W}$ without self loops. Following similar derivations, we obtain the optimal weight matrix $\mathbf{W}_*$ which makes $\mathcal{L}([\mathbf{A}\mathbf{X}]_{\mathcal{T}_\mathcal{V},:}\mathbf{W}_*, [\mathbf{Y}]_{\mathcal{T}_\mathcal{V},:}) = 0$ as:*

$$\mathbf{W}_* = ([\mathbf{A}\mathbf{X}]_{\mathcal{T}_S,:})^{-1} = \frac{1}{(1-h|\mathcal{Y}|)d} \begin{bmatrix} -(|\mathcal{Y}|-2+h) & 1-h & \cdots & 1-h \\ 1-h & -(|\mathcal{Y}|-2+h) & \cdots & 1-h \\ \vdots & \vdots & \ddots & \vdots \\ 1-h & 1-h & \cdots & -(|\mathcal{Y}|-2+h) \end{bmatrix} \tag{12}$$

*Again if for an arbitrary $(v, y_v) \in \mathcal{T}_\mathcal{V}$, a perturbation is added to the neighborhood $N(v)$ of the node $v$, such that the number of nodes with a randomly selected class label $y_p \in \mathcal{Y} \neq y_v$ is $\delta_2$ less than expected in $N(v)$, we have:*

$$[\mathbf{A}_\triangle \mathbf{X}]_{v,:} = \left[\begin{array}{ccccc} hd & \frac{1-h}{|\mathcal{Y}|-1}d - \delta_2 & \frac{1-h}{|\mathcal{Y}|-1}d & \cdots & \frac{1-h}{|\mathcal{Y}|-1}d \end{array}\right]$$

*Then applying the optimal weight matrix that we learned on $\mathcal{T}_\mathcal{V}$ to the aggregated feature on perturbed neighborhood $[\mathbf{A}_\triangle \mathbf{X}]_{v,:}$, we obtain $[\mathbf{A}_\triangle \mathbf{X}]_{v,:} \mathbf{W}_*$ which equals to:*

$$\left[\begin{array}{ccccc} 1 - \frac{(1-h)\delta_2}{(1-h|\mathcal{Y}|)d} & \frac{(|\mathcal{Y}|-2+h)\delta_2}{(1-h|\mathcal{Y}|)d} & -\frac{(1-h)\delta_2}{(1-h|\mathcal{Y}|)d} & \cdots & -\frac{(1-h)\delta_2}{(1-h|\mathcal{Y}|)d} \end{array}\right]$$

*Thus, the GCN layer formulated as $\mathbf{AXW}$ would misclassify when the following inequality holds:*

$$1 - \frac{(1-h)\delta_2}{(1-h|\mathcal{Y}|)d} < \frac{(|\mathcal{Y}|-2+h)\delta_2}{(1-h|\mathcal{Y}|)d}$$

*Or the amount of perturbation is:*

$$\begin{cases} \delta_2 > \frac{(1-h|\mathcal{Y}|)d}{|\mathcal{Y}|-1}, \text{when } 0 \leq h < \frac{1}{|\mathcal{Y}|} \\ \delta_2 < \frac{(1-h|\mathcal{Y}|)d}{|\mathcal{Y}|-1}, \text{when } h > \frac{1}{|\mathcal{Y}|} \end{cases} \tag{13}$$

*As a result, the least absolute amount of perturbation needed is $|\delta_2| = |\frac{(1-h|\mathcal{Y}|)d}{|\mathcal{Y}|-1}|$.*

*By comparing the least absolute amount of perturbation needed for both formulations to misclassify ($|\delta_1| = |\frac{-h|\mathcal{Y}|d-|\mathcal{Y}|+d+1}{|\mathcal{Y}|-1}|$ derived in Eq. (11) for the $(\mathbf{A}+\mathbf{I})\mathbf{XW}$ formulation; $|\delta_2| = |\frac{(1-h|\mathcal{Y}|)d}{|\mathcal{Y}|-1}|$ derived in Eq. (13) for the $\mathbf{AXW}$ formulation), we can see that $|\delta_1| = |\delta_2|$ if and only if $\delta_1 = -\delta_2$, which happens when $h = \frac{1-|\mathcal{Y}|+2d}{2|\mathcal{Y}|d}$. When $h < \frac{1-|\mathcal{Y}|+2d}{2|\mathcal{Y}|d}$ (heterophily), we have $|\delta_1| < |\delta_2|$, which means the $(\mathbf{A}+\mathbf{I})\mathbf{XW}$ formulation is less robust to perturbation than the $\mathbf{AXW}$ formulation. ∎*

**Discussions** From the above proof, we can see that the least absolute amount of perturbation $|\delta|$ needed for both GCN formulations is a function of the assumed homophily ratio $h$, the node degree $d$ for each node in the training set $\mathcal{T}_\mathcal{V}$, and the size of the class label set $|\mathcal{Y}|$. Fig. 4 shows the plots of $|\delta_1|$ and $|\delta_2|$ as functions of $h$, $|\mathcal{Y}|$ and $d$: from Fig. 4a, we can see that the least absolute amount of perturbations $|\delta|$ needed for both formulation first decreases as the assumed homophily level $h$ increases, until $\delta$ reaches 0, where the GCN layer predicts the same probability for all class labels; after that, $\delta$ decreases further below 0, and $|\delta|$ increases as $h$ increases; the $(\mathbf{A}+\mathbf{I})\mathbf{XW}$ formulation is less robust to perturbation than the $\mathbf{AXW}$ formulation at low homophily level until $h = \frac{1-|\mathcal{Y}|+2d}{2|\mathcal{Y}|d}$ as our proof shows, where $|\delta_1| = |\delta_2|$. Figure 4b shows the changes of $|\delta|$ as a function of $|\mathcal{Y}|$ when fixed $h = 0.1$ and $d = 20$. For both formulations, $|\delta|$ first decrease rapidly as $|\mathcal{Y}|$ increases until $\delta$ reaches 0, after that $\delta$ increases slowly as $|\mathcal{Y}|$ increases; this reveals that both GCN formulations are more robust when $|\mathcal{Y}| << d$ under high homophily level, and in that case $\mathbf{AXW}$ formulation is more robust than the $(\mathbf{A}+\mathbf{I})\mathbf{XW}$ formulation. Figure 4c shows the changes of $|\delta|$ as a function of $d$ for fixed $h = 0.1$ and $|\mathcal{Y}| = 5$: in this case the $\mathbf{AXW}$ formulation is always more robust than the $(\mathbf{A}+\mathbf{I})\mathbf{XW}$ formulation, and for the $(\mathbf{A}+\mathbf{I})\mathbf{XW}$ formulation, $|\delta|$ follows again a "V"-shape curve as $d$ changes.

### C.2 Detailed Analysis of Theorem 2

**Proof 2 (for Theorem 2)** *For all $v \in \mathcal{V}$, since its neighbors' class labels $\{y_u : u \in N(v)\}$ are conditionally independent given $y_v$, we can define a matrix $\mathbf{P}_v$ for each node $v$ as $[\mathbf{P}_v]_{i,j} = P(y_u = j|y_v = i), \forall i, j \in \mathcal{Y}, u \in N(v)$. Following the assumption that for all $v \in \mathcal{V}$, $P(y_u = y_v|y_v) = h$, $P(y_u = y|y_v) = \frac{1-h}{|\mathcal{Y}|-1}, \forall y \neq y_v$, we have*

$$\mathbf{P}_v = \mathbf{P} = \begin{bmatrix} h & \frac{1-h}{|\mathcal{Y}|-1} & \cdots & \frac{1-h}{|\mathcal{Y}|-1} \\ \frac{1-h}{|\mathcal{Y}|-1} & h & \cdots & \frac{1-h}{|\mathcal{Y}|-1} \\ \vdots & \vdots & \ddots & \vdots \\ \frac{1-h}{|\mathcal{Y}|-1} & \frac{1-h}{|\mathcal{Y}|-1} & \cdots & h \end{bmatrix}, \forall v \in \mathcal{V} \tag{14}$$

(a) $|\delta|$ as a function of $h$ under $d = 20, |\mathcal{Y}| = 5$.

(b) $|\delta|$ as a function of $|\mathcal{Y}|$ under $h = 0.1, d = 20$.

(c) $|\delta|$ as a function of $d$ under $h = 0.1, |\mathcal{Y}| = 5$.

Figure 4: Perturbation $|\delta|$ needed in order for GCN layers $(\mathbf{A} + \mathbf{I})\mathbf{X}\mathbf{W}$ and $\mathbf{A}\mathbf{X}\mathbf{W}$ to misclassify a node: Examples of perturbation $|\delta|$ as functions of $h$, $|\mathcal{Y}|$ and $d$, respectively.

*Now consider node $w \in N_2(v)$, we have:*

$$P(y_w = k | y_v = i) = \sum_{j \in |\mathcal{Y}|} P(y_w = k | y_u = j) P(y_u = j | y_v = i) = \sum_{j \in |\mathcal{Y}|} [\mathbf{P}]_{j,k}[\mathbf{P}]_{i,j} = \mathbf{P}^2 \quad (15)$$

*Therefore, to prove that the 2-hop neighborhood $N_2(v)$ for any node $v \in \mathcal{V}$ is homophily-dominant in expectation (i.e. $P(y_w = i | y_v = i) \geq P(y_w = j | y_v = i), \forall j \in \mathcal{Y} \neq i, w \in N_2(v))$, we need to show that the diagonal entries $[\mathbf{P}^2]_{i,i}$ of $\mathbf{P}^2$ are larger than the off-diagonal entries $[\mathbf{P}^2]_{i,j}$.*

*Denote $\rho = \frac{1-h}{|\mathcal{Y}|-1}$. From Eq. (14), we have*

$$[\mathbf{P}^2]_{i,i} = h^2 + (|\mathcal{Y}| - 1)\rho^2 \quad (16)$$

*and for $i \neq j$*

$$[\mathbf{P}^2]_{i,j} = 2h\rho + (|\mathcal{Y}| - 2)\rho^2 \quad (17)$$

*Thus,*

$$[\mathbf{P}^2]_{i,i} - [\mathbf{P}^2]_{i,j} = h^2 - 2h\rho + \rho^2 = (h - \rho)^2 \geq 0$$

*with equality if and only if $h = \rho$, namely $h = \frac{1}{|\mathcal{Y}|}$. Therefore, we proved that the 2-hop neighborhood $N_2(v)$ for any node $v \in \mathcal{V}$ will always be homophily-dominant in expectation.* ∎

### C.3 Detailed Analysis of Theorem 3

**Preliminaries** We define unnormalized Laplacian matrix of graph $\mathcal{G}$ as $\mathbf{L} = \mathbf{D} - \mathbf{A}$, where $\mathbf{A} \in \{0, 1\}^{|\mathcal{V}| \times |\mathcal{V}|}$ is the adjacency matrix and $\mathbf{D}$ is the diagonal matrix with $[\mathbf{D}]_{i,i} = \sum_j [\mathbf{A}]_{i,j}$. Without loss of generality, since the eigenvalues $\{\lambda_i\}$ of $\mathbf{L}$ are real and nonnegative [32], we assume the following order for the eigenvalues of $\mathbf{L}$: $0 = \lambda_0 < \lambda_1 \leq \lambda_2 \leq \cdots \leq \lambda_{|\mathcal{V}|-1} = \lambda_{max}$. Furthermore, since $\mathbf{L}$ is real and symmetric, there exists a set of orthonormal eigenvectors $\{\mathbf{v}_i\}$ that form a complete basis of $\mathbb{R}^{|\mathcal{V}|}$. This means that for any graph signal $\mathbf{s} \in \mathbb{R}^{|\mathcal{V}|}$, where $\mathbf{s}_u$ is the value of the signal on node $u \in \mathcal{V}$, it can be decomposed to a weighted sum of $\{\mathbf{v}_i\}$. Mathematically, $\mathbf{s}$ is represented as $\mathbf{s} = \sum_{i=0}^{|\mathcal{V}|-1} c_{s,i}\mathbf{v}_i$, where $c_{s,i} = \mathbf{s}^\mathsf{T}\mathbf{v}_i$. We regard $c_{s,i}$ as the coefficient of $\mathbf{s}$ at frequency component $i$ and regard the coefficients at all frequencies components $\{c_{s,i}\}$ as the spectrum of signal $\mathbf{s}$ with respect to graph $\mathcal{G}$. In the above order of the eigenvalues, $\lambda_i$ which are closer to 0 would correspond to lower-frequency components, and $\lambda_i$ which are closer to $\lambda_{max}$ would correspond to higher-frequency components. Interested readers are referred to [32] for further details regarding signal processing on graphs.

The **smoothness score of a signal $\mathbf{s}$** on graph $\mathcal{G}$, which measures the amount of changes of signal $\mathbf{s}$ along the edges of graph $\mathcal{G}$, can be defined using $\mathbf{L}$ as

$$\mathbf{s}^\mathsf{T}\mathbf{L}\mathbf{s} = \sum_{i,j} \mathbf{A}_{ij}(\mathbf{s}_i - \mathbf{s}_j)^2 = \sum_{u \in \mathcal{V}} \sum_{v \in N(u)} (\mathbf{s}_u - \mathbf{s}_v)^2. \quad (18)$$

Then, for two eigenvectors $\mathbf{v}_i$ and $\mathbf{v}_j$ corresponding to eigenvalues $\lambda_i \leq \lambda_j$ of $\mathbf{L}$, we have:

$$\mathbf{v}_i^{\mathsf{T}}\mathbf{L}\mathbf{v}_i = \lambda_i \leq \lambda_j = \mathbf{v}_j^{\mathsf{T}}\mathbf{L}\mathbf{v}_j$$

which means that $\mathbf{v}_i$ is *more smooth* than $\mathbf{v}_j$. This matches our expectations that a lower-frequency signal on $\mathcal{G}$ should have smaller smoothness score. The **smoothness score for arbitrary graph signal** $\mathbf{s} \in \mathbb{R}^{|\mathcal{V}|}$ can be represented by its coefficients of each frequency component as:

$$\mathbf{s}^{\mathsf{T}}\mathbf{L}\mathbf{s} = \left( \sum_i c_{s,i}\mathbf{v}_i \right) \mathbf{L} \left( \sum_i c_{s,i}\mathbf{v}_i \right) = \sum_{i=0}^{|\mathcal{V}|-1} c_{s,i}^2 \lambda_i \tag{19}$$

with the above preliminaries, we can define the following concept:

**Definition 5** *Suppose* $\mathbf{s} = \sum_{i=0}^{|\mathcal{V}|-1} c_{s,i}\mathbf{v}_i$ *and* $\mathbf{t} = \sum_{i=0}^{|\mathcal{V}|-1} c_{t,i}\mathbf{v}_i$ *are two graph signals defined on* $\mathcal{G}$. *In the spectrum of the unnormalized graph laplacian* $\mathbf{L}$, *graph signal* $\mathbf{s}$ *has higher energy on high-frequency components than* $\mathbf{t}$ *if there exists integer* $0 < M \leq |\mathcal{V}| - 1$ *such that* $\sum_{i=M}^{|\mathcal{V}|-1} c_{s,i}^2 > \sum_{i=M}^{|\mathcal{V}|-1} c_{t,i}^2$.

Based on these preliminary definitions, we can now proceed with the proof of the theorem:

**Proof 3 (for Theorem 3)** *We first prove that for graph signals* $\mathbf{s}, \mathbf{t} \in \{0, 1\}^{|\mathcal{V}|}$, *edge homophily ratio* $h_s < h_t$ *if and only if* $\mathbf{s}^{\mathsf{T}}\mathbf{L}\mathbf{s} > \mathbf{t}^{\mathsf{T}}\mathbf{L}\mathbf{t}$. *Following Dfn. 1, the edge homophily ratio for signal* $\mathbf{s}$ *(similarly for* $\mathbf{t}$*) can be calculated as:*

$$h_s = \frac{1}{2|\mathcal{E}|} \sum_{u \in \mathcal{V}} \left( d_u - \sum_{v \in N(v)} (\mathbf{s}_u - \mathbf{s}_v)^2 \right) = \frac{1}{2|\mathcal{E}|} \sum_{u \in \mathcal{V}} d_u - \frac{1}{2|\mathcal{E}|} \sum_{u \in \mathcal{V}} \sum_{v \in N(v)} (\mathbf{s}_u - \mathbf{s}_v)^2 \tag{20}$$

*Plugging this in Eq. (18), we obtain:*

$$h_s = \frac{1}{2|\mathcal{E}|} \sum_{u \in \mathcal{V}} d_u - \frac{1}{2|\mathcal{E}|}\mathbf{s}^{\mathsf{T}}\mathbf{L}\mathbf{s} = 1 - \frac{1}{2|\mathcal{E}|}\mathbf{s}^{\mathsf{T}}\mathbf{L}\mathbf{s}$$

*where* $|\mathcal{E}|$ *is the number of edges in* $\mathcal{G}$. *From the above equation, we have*

$$h_s < h_t \iff 1 - \frac{1}{2|\mathcal{E}|}\mathbf{s}^{\mathsf{T}}\mathbf{L}\mathbf{s} < 1 - \frac{1}{2|\mathcal{E}|}\mathbf{t}^{\mathsf{T}}\mathbf{L}\mathbf{t} \iff \mathbf{s}^{\mathsf{T}}\mathbf{L}\mathbf{s} > \mathbf{t}^{\mathsf{T}}\mathbf{L}\mathbf{t}$$

*i.e. edge homophily ratio* $h_s < h_t$ *if and only if* $\mathbf{s}^{\mathsf{T}}\mathbf{L}\mathbf{s} > \mathbf{t}^{\mathsf{T}}\mathbf{L}\mathbf{t}$.

*Next we prove that if* $\mathbf{s}^{\mathsf{T}}\mathbf{L}\mathbf{s} > \mathbf{t}^{\mathsf{T}}\mathbf{L}\mathbf{t}$, *then following Dfn.5, signal* $\mathbf{s}$ *has higher energy on high-frequency components than* $\mathbf{t}$. *We prove this by contradiction: suppose integer* $0 < M \leq |\mathcal{V}| - 1$ *does not exist such that* $\sum_{i=M}^{|\mathcal{V}|-1} c_{si}^2 > \sum_{i=M}^{|\mathcal{V}|-1} c_{ti}^2$ *when* $\mathbf{s}^{\mathsf{T}}\mathbf{L}\mathbf{s} > \mathbf{t}^{\mathsf{T}}\mathbf{L}\mathbf{t}$, *then all of the following inequalities must hold, as the eigenvalues of* $\mathbf{L}$ *satisfy* $0 = \lambda_0 < \lambda_1 \leq \lambda_2 \leq \cdots \leq \lambda_{|\mathcal{V}|-1} = \lambda_{max}$:

$$0 = \lambda_0(c_{s,0}^2 + c_{s,1}^2 + c_{s,2}^2 + \cdots + c_{s,|\mathcal{V}|-1}^2) = \lambda_0(c_{t,0}^2 + c_{t,1}^2 + c_{t,2}^2 + \cdots + c_{t,|\mathcal{V}|-1}^2) = 0$$

$$(\lambda_1 - \lambda_0)(c_{s,1}^2 + c_{s,2}^2 + \cdots + c_{s,|\mathcal{V}|-1}^2) \leq (\lambda_1 - \lambda_0)(c_{t,1}^2 + c_{t,2}^2 + \cdots + c_{t,|\mathcal{V}|-1}^2)$$

$$(\lambda_2 - \lambda_1)(c_{s,2}^2 + \cdots + c_{s,|\mathcal{V}|-1}^2) \leq (\lambda_2 - \lambda_1)(c_{t,2}^2 + \cdots + c_{t,|\mathcal{V}|-1}^2)$$

$$\vdots$$

$$(\lambda_{|\mathcal{V}|-1} - \lambda_{|\mathcal{V}|-2})c_{s,|\mathcal{V}|-1}^2 \leq (\lambda_{|\mathcal{V}|-1} - \lambda_{|\mathcal{V}|-2})c_{t,|\mathcal{V}|-1}^2$$

*Summing over both sides of all the above inequalities, we have*

$$\lambda_0 \cdot c_{s,0}^2 + \lambda_1 \cdot c_{s,1}^2 + \lambda_2 \cdot c_{s,2}^2 + \cdots + \lambda_{|\mathcal{V}|-1} \cdot c_{s,|\mathcal{V}|-1}^2 \leq \lambda_0 \cdot c_{t,0}^2 + \lambda_1 \cdot c_{t,1}^2 + \lambda_2 \cdot c_{t,2}^2 + \cdots + \lambda_{|\mathcal{V}|-1} \cdot c_{t,|\mathcal{V}|-1}^2$$

*i.e.,* $\sum_{i=0}^{|\mathcal{V}|-1} c_{si}^2 \lambda_i \leq \sum_{i=0}^{|\mathcal{V}|-1} c_{ti}^2 \lambda_i$. *However, from Eq. (19), we should have*

$$\mathbf{s}^{\mathsf{T}}\mathbf{L}\mathbf{s} > \mathbf{t}^{\mathsf{T}}\mathbf{L}\mathbf{t} \iff \sum_{i=0}^{|\mathcal{V}|-1} c_{si}^2 \lambda_i > \sum_{i=0}^{|\mathcal{V}|-1} c_{ti}^2 \lambda_i$$

*which contradicts with the previous resulting inequality. Therefore, the assumption should not hold, and there must exist an integer* $0 < M \leq |\mathcal{V}| - 1$ *such that* $\sum_{i=M}^{|\mathcal{V}|-1} c_{si}^2 > \sum_{i=M}^{|\mathcal{V}|-1} c_{ti}^2$ *when* $\mathbf{s}^{\mathsf{T}}\mathbf{L}\mathbf{s} > \mathbf{t}^{\mathsf{T}}\mathbf{L}\mathbf{t}$. ∎

**Extension of Theorem 3 to one-hot encoding of class label vectors** Theorem 3 discusses only the graph signal $\mathbf{s}, \mathbf{t} \in \{0,1\}^{|\mathcal{V}|}$ with only 1 channel (i.e., with only 1 value assigned to each node). It is possible to generalize the theorem to one-hot encoding $\mathbf{Y}_s, \mathbf{Y}_t \in \{0,1\}^{|\mathcal{V}| \times |\mathcal{Y}|}$ as graph signal with $|\mathcal{Y}|$-channels by modifying Dfn. 5 as follows:

**Definition 6** *Suppose* $[\mathbf{Y}_s]_{:,j} = \sum_{i=0}^{|\mathcal{V}|-1} c_{s,j,i} \mathbf{v}_i$ *and* $[\mathbf{Y}_t]_{:,j} = \sum_{i=0}^{|\mathcal{V}|-1} c_{t,j,i} \mathbf{v}_i$ *are one-hot encoding of class label vector* $\mathbf{y}_s, \mathbf{y}_t$ *defined as graph signals on* $\mathcal{G}$*, where* $c_{s,j,i} = [\mathbf{Y}_s]_{:,j}^\mathsf{T} \mathbf{v}_i$ *is the coefficient of the jth-channel of* $\mathbf{Y}_s$ *at frequency component* $i$*. In the spectrum of the unnormalized graph laplacian* $\mathbf{L}$*, graph signal* $\mathbf{Y}_s$ *has higher energy on high-frequency components than* $\mathbf{Y}_t$ *if there exists integer* $0 < M \leq |\mathcal{V}| - 1$ *such that* $\sum_{i=M}^{|\mathcal{V}|-1} \sum_{j=1}^{\phi} c_{s,j,i}^2 > \sum_{i=M}^{|\mathcal{V}|-1} \sum_{j=1}^{\phi} c_{t,j,i}^2$*.*

Under this definition, we can prove Theorem 3 for one-hot encoding of class label vectors $\mathbf{Y}_s, \mathbf{Y}_t$ as before, with the modification that in this case we have for signal $\mathbf{Y}_s$ (similarly for $\mathbf{Y}_t$):

$$h_s = \frac{1}{4|\mathcal{E}|} \sum_{u \in \mathcal{V}} \left( 2d_u - \sum_{v \in N(v)} \sum_{j=1}^{\phi} ([\mathbf{Y}_s]_{u,j} - [\mathbf{Y}_t]_{v,j})^2 \right)$$

instead of Eq. (20). The rest of the proof is similar to Proof 3.

# D   Our H$_2$GCN model: Details

In this section, we give the pipeline and pseudocode of H$_2$GCN, elaborate on its differences from existing GNN models, and present a detailed analysis of its computational complexity.

## D.1   Pseudocode & Pipeline

In Fig. 5 we visualize H$_2$GCN, which we describe in § 3.2. We also give its pseudocode in Algorithm 1.

## D.2   Detailed Comparison of H$_2$GCN to existing GNN models

In § 4, we discussed several high-level differences between H$_2$GCN and the various GNN models that we consider in this work, including the inclusion or not of designs D1-D3. Here we give some additional conceptual and mechanism differences.

As we have mentioned, H$_2$GCN differs from GCN [17] in a number of ways: (1) In each round of propagation/aggregation, GCN "mixes" the ego- and neighbor-representations by repeatedly averaging them to obtain the new node representations, while H$_2$GCN keeps them distinct via concatenation; (2) GCN considers only the 1-hop neighbors (including the ego / self-loops), while H$_2$GCN considers higher-order neighborhoods ($\bar{N}_1$ and $\bar{N}_2$); (3) GCN applies non-linear embedding transformations per round (e.g., RELU), while H$_2$GCN perform feature embedding for the ego in the first layer and drops all other non-linearities in the aggregation stage; and (4) GCN does not use the jumping knowledge framework (unlike H$_2$GCN), and makes the node classification predictions based on the last-round representations.

Unlike GAT, H$_2$GCN does not use any attention mechanism. Creating attention mechanisms that can generalize well to heterophily is an interesting future direction. Moreover, GCN-Cheby uses entirely different mechanisms than the other GNN models that we consider (i.e., Chebysev polynomials), though it has some conceptual similarities to H$_2$GCN in terms of the higher-order neighborhoods that it models.

GraphSAGE differs from H$_2$GCN in the same ways that are described in (2)-(4) above. In addition to leveraging only the 1-hop neighborhood, GraphSAGE also samples a fixed number of neighbors per round, while H$_2$GCN uses the full neighborhood. With respect to ego- and neighbor-representations, GraphSAGE concatenates them (as we do) but subsequently applies non-linear embedding transformations to them jointly (while we simplify all non-linear transformations). Our empirical analysis has revealed that such transformations lead to a decrease in performance in heterophily settings (see paragraph below on "Non-linear embedding transformations...").

Figure 5: H$_2$GCN-2 pipeline. It consists of 3 stages: **(S1)** feature embedding, **(S2)** neighborhood aggregation, and **(S3)** classification. The *feature embedding stage* **(S1)** uses a graph-agnostic dense layer to generate the feature embedding $\mathbf{r}_v^{(0)}$ of each node $v$ based on its ego-feature $\mathbf{x}_v$. In the *neighborhood aggregation stage* **(S2)**, the generated embeddings are aggregated and repeatedly updated within the node's neighborhood; the 1-hop neighbors $N_1(v)$ and 2-hop neighbors $N_2(v)$ are aggregated separately and then concatenated, following our design D2. In the *classification stage* **(S3)**, each node is classified based on its final embedding $\mathbf{r}_v^{(\text{final})}$, which consists of its intermediate representations concatenated as per design D3.

Finally, MixHop differs from H$_2$GCN in the same ways that are described in (1) and (3)-(4) above. It explicitly considers higher-order neighborhoods up to $N_2$, though [1] defines the 2-hop neighborhoods as that including neighbors *up to* 2-hop away neighbors. In our framework, we define the $i$-hop neighborhood as the set of neighbors with minimum distance exactly $i$ from the ego (§ 2). Finally, the output layer of MixHop uses a tailored, column-wise attention layer, which prioritizes specific features, before the softmax layer. In contrast, before the classification layer, H$_2$GCN uses concatenation-based jumping knowledge in order to represent the high-frequency components that are critical in heterophily.

**Non-linear embedding transformations per round in H$_2$GCN?** GCN [17], GraphSAGE [11] and other GNN models embed the intermediate representations per round of feature propagation and aggregation. However, as we show in the ablation study in App. G.2 (Table G.4, last row "Non-linear"), introducing non-linear transformations per round of the neighborhood aggregation stage **(S2)** of H$_2$GCN-2 (i.e., with $K = 2$) as follows leads to *worse* performance than the framework design that we introduce in Eq. (5) of § 3.2:

$$\mathbf{r}_v^{(k)} = \texttt{COMBINE}\left(\sigma\left(\mathbf{W}\left[\mathbf{r}_v^{(k-1)}, \texttt{AGGR}\{\mathbf{r}_u^{(k-1)} : u \in N_1(v)\}, \texttt{AGGR}\{\mathbf{r}_u^{(k-1)} : u \in N_2(v)\}\right]\right)\right), \quad (21)$$

where $\sigma$ is RELU and $\mathbf{W}$ is a learnable matrix. Our design in Eq. 5 aggregates different neighborhoods in a similar way to SGC [37], which has shown that removing non-linearities does not negatively impact performance in homophily settings. We actually find that removing non-linearities even *improves* the performance under heterophily.

### D.3 H$_2$GCN: Time Complexity in Detail

**Preliminaries** The worst case time complexity for calculating $\mathbf{A} \cdot \mathbf{B}$ when both $\mathbf{A}$ and $\mathbf{B}$ are sparse matrices is $\text{O}(\text{nnz}(\mathbf{A}) \cdot c_{\mathbf{B}})$, where $\text{nnz}(\mathbf{A})$ is the number of non-zero elements in matrix

**Algorithm 1:** $H_2$GCN Framework for Node Classification under Homophily & Heterophily

---

**Input:** Graph Adjacency Matrix $\mathbf{A} \in \{0,1\}^{n \times n}$; Node Feature Matrix $\mathbf{X} \in \mathbb{R}^{n \times F}$; Set of Labels $\mathcal{Y}$;
      Labeled Nodes $\mathcal{T}_\mathcal{V}$
**Hyper-parameters:** Dropout Rate; Non-linearity function $\sigma$; Number of Embedding Rounds $K$;
                Dimension of Feature Embedding $p$;
**Network Parameters:** $\mathbf{W}_e \in \mathbb{R}^{F \times p}$; $\mathbf{W}_c \in \mathbb{R}^{(2^{K+1}-1)p \times |\mathcal{Y}|}$
**Output:** Class label vector $\mathbf{y}$
**begin**

    /* All new variables defined below are initialized as all 0 */

    /* Stage S1: Feature Embedding */
    **for** $v \in \mathcal{V}$ **do**
        $\mathbf{r}_v^{(0)} \leftarrow \sigma\left(\mathbf{x}_v \mathbf{W}_e\right)$ /* Embeddings stored in matrix $\mathbf{R}$ */

    /* Stage S2: Neighborhood Aggregation */
    /* Calculate higher-order neighborhoods $\bar{N}_1$ and $\bar{N}_2$ without self-loops and
       their corresponding adjacency matrices $\bar{\mathbf{A}}_1$ and $\bar{\mathbf{A}}_2$ */
    $\mathbf{A}_0 \leftarrow \mathbf{I}_n$                             /* $\mathbf{I}_n$ is the $n \times n$ identity matrix */
    $\bar{\mathbf{A}}_1 \leftarrow \mathbb{I}\left[\mathbf{A} - \mathbf{I}_n > 0\right]$     /* $\mathbb{I}$ is a element-wise indicator function for matrix */
    $\bar{\mathbf{A}}_2 \leftarrow \mathbb{I}\left[\mathbf{A}^2 - \mathbf{A} - \mathbf{I}_n > 0\right]$;
    **for** $i \leftarrow 1$ **to** $2$ **do**
        **for** $v \in \mathcal{V}$ **do**
            $d_{v,i} \leftarrow \sum_k \bar{a}_{vk,i}$              /* degree of node $v$ at neighborhood $\bar{N}_i$ */
        $\bar{\mathbf{D}}_i \leftarrow \text{diag}\{d_{v,i} : v \in \mathcal{V}\}$;
        $\bar{\mathbf{A}}_i \leftarrow \bar{\mathbf{D}}_i^{-\frac{1}{2}} \bar{\mathbf{A}}_i \bar{\mathbf{D}}_i^{-\frac{1}{2}}$      /* symmetric degree-normalization of matrices $\bar{\mathbf{A}}_i$ */
    **for** $k \leftarrow 1$ **to** $K$ **do**
        $\mathbf{R}_1^{(k)} \leftarrow \bar{\mathbf{A}}_1 \mathbf{R}^{(k-1)}$                    /* Designs D1 + D2 */
        $\mathbf{R}_2^{(k)} \leftarrow \bar{\mathbf{A}}_2 \mathbf{R}^{(k-1)}$;
        /* $\|$ is the vector concatenation operator */
        $\mathbf{R}^{(k)} \leftarrow \left(\mathbf{R}_1^{(k)} \| \mathbf{R}_2^{(k)}\right)$
    $\mathbf{R}^{(\text{final})} \leftarrow \left(\mathbf{R}^{(0)} \| \mathbf{R}^{(1)} \| \ldots \| \mathbf{R}^{(K)}\right)$            /* Design D3 */

    /* Stage S3: Classification */
    $\mathbf{R}^{(\text{final})} \leftarrow \text{dropout}(\mathbf{R}^{(\text{final})})$           /* default dropout rate: 0.5 */
    **for** $v \in \mathcal{V}$ **do**
        $\mathbf{p}_v \leftarrow \text{softmax}(\mathbf{r}_v^{(\text{final})} \mathbf{W}_c)$;
        $\mathbf{y}_v \leftarrow \arg\max(\mathbf{p}_v)$                /* class label */

---

$\mathbf{A}$, and $c_\mathbf{B} = \max(\sum_j \mathbb{I}[b_{ij} > 0])$ is the maximum number of non-zero elements in any row of matrix $\mathbf{B}$. The time complexity for calculating $\mathbf{A} \cdot \mathbf{X}$, when $\mathbf{X}$ is a dense matrix with $F$ columns, is $O(\text{nnz}(\mathbf{A})F)$.

**Time complexity of $H_2$GCN**   We analyze the time complexity of $H_2$GCN by stage (except the classification stage).

The feature embedding stage **(S1)** takes $O(\text{nnz}(\mathbf{X})p)$ to calculate $\sigma(\mathbf{X}\mathbf{W}_e)$ where $\mathbf{W}_e \in \mathbb{R}^{F \times p}$ is a learnable dense weight matrix, and $\mathbf{X} \in \mathbb{R}^{n \times F}$ is the node feature matrix.

In the neighborhood aggregation stage **(S2)**, we perform the following computations:

- *Calculation of higher-order neighborhoods.* Given that $\mathbf{A}$ is sparse, we can obtain the 2-hop neighborhood by calculating $\mathbf{A}^2$ in $O\left(|\mathcal{E}|d_{\max}\right)$, where $|\mathcal{E}|$ is the number of edges in $\mathcal{G}$ (equal to the number of non-zeroes in $\mathbf{A}$), and $d_{\max}$ is the maximum degree across all nodes $v \in \mathcal{V}$ (which is equal to the maximum number of non-zeroes in any row of $\mathbf{A}$).

- *Feature Aggregation.* We begin with a $p$-dimensional embedding for each node after feature embedding. In round $k$, since we are using the neighborhoods $\bar{N}_1$ and $\bar{N}_2$, we have an em-

bedding $\mathbf{R}^{(k-1)} \in \mathbb{R}^{n \times 2^{(k-1)}p}$ as input. We aggregate embedding vectors within neighborhood by $\mathbf{R}^{(k)} = \left(\bar{\mathbf{A}}_1 \mathbf{R}^{(k-1)} \| \bar{\mathbf{A}}_2 \mathbf{R}^{(k-1)}\right)$, in which $\bar{\mathbf{A}}_i$ corresponds to the adjacency matrix of neighborhood $\bar{N}_i$. The two sparse matrix-matrix multiplications in the concatenation take $\mathrm{O}\left(|\mathcal{E}|2^{(k-1)}p + |\mathcal{E}_2|2^{(k-1)}p\right)$, where $|\mathcal{E}_2| = \frac{1}{2}\sum_{v \in \mathcal{V}}|\bar{N}_2(v)|$. Over $K$ rounds of embedding, the complexity becomes $\mathrm{O}\left(2^K(|\mathcal{E}| + |\mathcal{E}_2|)p\right)$.

Adding all the big-O terms above, we have the overall time complexity for stages **(S1)** and **(S2)** of $\mathrm{H}_2\mathrm{GCN}$ as:

$$\mathrm{O}\left(\mathrm{nnz}(\mathbf{X})\,p + |\mathcal{E}|d_{\max} + 2^K(|\mathcal{E}| + |\mathcal{E}_2|)p\right),$$

where $K$ is usually a small number (e.g., 2). For small values of $K$, the complexity becomes $\mathrm{O}\left(|\mathcal{E}|d_{\max} + (\mathrm{nnz}(\mathbf{X}) + |\mathcal{E}| + |\mathcal{E}_2|)p\right)$.

# E  Additional Related Work

In § 4, we discuss relevant work on GNNs. Here we briefly mention other approaches for node classification.

Collective classification in statistical relational learning focuses on the problem of node classification by leveraging the correlations between the node labels and their attributes [30]. Since exact inference is NP-hard, approximate inference algorithms (e.g., iterative classification [14, 20], loopy belief propagation) are used to solve the problem. Belief propagation (BP) [40] is a classic message-passing algorithm for graph-based semi-supervised learning, which can be used for graphs exhibiting homophily or heterophily [19] and has fast linearized versions [10, 8]. Different from the setup where GNNs are employed, BP does *not* by itself leverage node features, and usually assumes a pre-defined class compatibility or edge potential matrix (§ 2). We note, however, that Gatterbauer [9] proposed estimating the class compatibility matrix instead of using a pre-defined one in the BP formulation. Moreover, the recent CPGNN model [43] integrates the compatibility matrix as a set of learnable parameters into GNN, which it initializes with an estimated class compatibility matrix. Another classic approach for collective classification or graph-based semi-supervised learning is label propagation, which iteratively propagates the (up-to-date) label information of each node to its neighbors in order to minimize the overall smoothness penalty of label assignments in the graph. Standard label propagation approaches inherently assume homophily by penalizing different label assignments among immediate neighborhoods, but more recent works have also looked into formulations which can better address heterophily: Before applying label propagation, Peel [25] transforms the original graph into either a similarity graph by measuring similarity between node neighborhoods or a new graph connecting nodes that are two hops away; Chin et al. [6] decouple graph smoothing where the notion of "identity" and "preference" for each node are considered separately. However, like BP, these approaches do not by themselves utilize node features.

# F  Experimental Setup & Hyperparameter Tuning

## F.1  Setup

**$\mathrm{H}_2\mathrm{GCN}$ Implementation**  We use $K = 1$ for $\mathrm{H}_2\mathrm{GCN}$-1 and $K = 2$ for $\mathrm{H}_2\mathrm{GCN}$-2. For loss function, we calculate the cross entropy between the predicted and the ground-truth labels for nodes within the training set, and add $L_2$ regularization of network parameters $\mathbf{W}_e$ and $\mathbf{W}_c$. (cf. Alg. 1)

**Baseline Implementations**  For all baselines besides MLP, we used the official implementation released by the authors on GitHub.

- **GCN & GCN-Cheby** [17]: `https://github.com/tkipf/gcn`
- **GraphSAGE** [11]: `https://github.com/williamleif/graphsage-simple` (PyTorch implementation)
- **MixHop** [1]: `https://github.com/samihaija/mixhop`
- **GAT** [36]: `https://github.com/PetarV-/GAT`. (For large datasets, we make use of the sparse version provided by the author.)

For MLP, we used our own implementation of MLP with 1-hidden layer, which is equivalent to the case of $K = 0$ in Algorithm 1. We use the same loss function as H$_2$GCN for training MLP.

**Hardware Specifications**   We run experiments on synthetic benchmarks with an Amazon EC2 instance with instance size as `p3.2xlarge`, which features an 8-core CPU, 61 GB Memory, and a Tesla V100 GPU with 16 GB GPU Memory. For experiments on real benchmarks, we use a workstation with a 12-core AMD Ryzen 9 3900X CPU, 64GB RAM, and a Quadro P6000 GPU with 24 GB GPU Memory.

### F.2   Tuning the GNN Models

To avoid bias, we tuned the hyperparameters of each method (H$_2$GCN and baseline models) on each benchmark. Below we list the hyperparameters tested on each benchmark per model. As the hyperparameters defined by each baseline model differ significantly, we list the combinations of non-default command line arguments we tested, without explaining them in detail. We refer the interested reader to the corresponding original implementations for further details on the arguments, including their definitions.

**Synthetic Benchmark Tuning**   For each synthetic benchmark, we report the results for different heterophily levels under the same set of hyperparameters for each method, so that we can compare how the same hyperparameters perform across the full spectrum of low-to-high homophily. We report the best performance, for the set of hyperparameters which performs the best on the validation set on the majority of the heterophily levels for each method.

For `syn-cora`, we test the following command-line arguments for each baseline method:

- **H$_2$GCN-1 & H$_2$GCN-2**:
  - Dimension of Feature Embedding $p$: 64
  - Non-linearity Function $\sigma$: ReLU
  - Dropout Rate: $a \in \{0, 0.5\}$

  We report the best performance, for $a = 0$.
- **GCN** [17]:
  - `hidden1`: $a \in \{16, 32, 64\}$
  - `early_stopping`: $b \in \{40, 100, 200\}$
  - `epochs`: 2000

  We report the best performance, for $a = 32, b = 40$.
- **GCN-Cheby** [17]:
  - Set 1:
    * `hidden1`: $a \in \{16, 32, 64\}$
    * `dropout`: 0.6
    * `weight_decay`: $b \in \{$`1e-5, 5e-4`$\}$
    * `max_degree`: 2
    * `early_stopping`: 40
  - Set 2:
    * `hidden1`: $a \in \{16, 32, 64\}$
    * `dropout`: 0.5
    * `weight_decay`: `5e-4`
    * `max_degree`: 3
    * `early_stopping`: 40

  We report the best performance, for Set 1 with $a = 64, b =$ `5e-4`.
- **GraphSAGE** [11]:
  - `hid_units`: $a \in \{64, 128\}$
  - `lr`: $b \in \{0.1, 0.7\}$
  - `epochs`: 500

We report the performance with $a = 64, b = 0.7$.

- **MixHop** [1]:
  - `hidden_dims_csv`: $a \in \{64, 192\}$
  - `adj_pows`: 0, 1, 2

  We report the performance with $a = 192$.
- **GAT** [36]:
  - `hid_units`: $a \in \{8, 16, 32, 64\}$
  - `n_heads`: $b \in \{1, 4, 8\}$

  We report the performance with $a = 8, b = 8$.
- **MLP**
  - Dimension of Feature Embedding $p$: 64
  - Non-linearity Function $\sigma$: ReLU
  - Dropout Rate: 0.5

For `syn-products`, we test the following command-line arguments for each baseline method:

- **H$_2$GCN-1 & H$_2$GCN-2**:
  - Dimension of Feature Embedding $p$: 64
  - Non-linearity Function $\sigma$: ReLU
  - Dropout Rate: $a \in \{0, 0.5\}$

  We report the best performance, for $a = 0.5$.
- **GCN** [17]:
  - `hidden1`: 64
  - `early_stopping`: $a \in \{40, 100, 200\}$
  - `epochs`: 2000

  In addition, we disabled the default feature normalization in the official implementation, as the feature vectors in this benchmark have already been normalized, and we found the default normalization method hurts the performance significantly. We report the best performance, for $a = 40$.
- **GCN-Cheby** [17]:
  - `hidden1`: 64
  - `max_degree`: 2
  - `early_stopping`: 40
  - `epochs`: 2000

  We also disabled the default feature normalization in the official implementation for this baseline.
- **GraphSAGE** [11]:
  - `hid_units`: $a \in \{64, 128\}$
  - `lr`: $b \in \{0.1, 0.7\}$
  - `epochs`: 500

  We report the performance with $a = 128, b = 0.1$.
- **MixHop** [1]:
  - `hidden_dims_csv`: $a \in \{64, 192\}$
  - `adj_pows`: 0, 1, 2

  We report the performance with $a = 192$.
- **GAT** [36]:
  - `hid_units`: 8

  We also disabled the default feature normalization in the official implementation for this baseline.
- **MLP**
  - Dimension of Feature Embedding $p$: 64
  - Non-linearity Function $\sigma$: ReLU
  - Dropout Rate: 0.5

**Real Benchmark (except Cora-Full) Tuning**    For each real benchmark in Table 5 (except **Cora-Full**), we perform hyperparameter tuning (see values below) and report the best performance of each method on the validation set. So, for each method, its performance on different benchmarks can be reported from different hyperparameters. We test the following command-line arguments for each baseline method:

- **$H_2$GCN-1 & $H_2$GCN-2**:
  - Dimension of Feature Embedding $p$: 64
  - Non-linearity Function $\sigma$: {ReLU, None}
  - Dropout Rate: $\{0, 0.5\}$
  - L2 Regularization Weight: {1e-5, 5e-4}
- **GCN** [17]:
  - hidden1: 64
  - early_stopping: $\{40, 100, 200\}$
  - epochs: 2000
- **GCN-Cheby** [17]:
  - hidden1: 64
  - weight_decay: {1e-5, 5e-4}
  - max_degree: 2
  - early_stopping: $\{40, 100, 200\}$
  - epochs: 2000
- **GraphSAGE** [11]:
  - hid_units: 64
  - lr: $\{0.1, 0.7\}$
  - epochs: 500
- **MixHop** [1]:
  - hidden_dims_csv: $\{64, 192\}$
  - adj_pows: 0, 1, 2
- **GAT** [36]:
  - hid_units: $8$
- **MLP**
  - Dimension of Feature Embedding $p$: 64
  - Non-linearity Function $\sigma$: {ReLU, None}
  - Dropout Rate: $\{0, 0.5\}$

For **GCN+JK**, **GCN-Cheby+JK** and **GraphSAGE+JK**, we enhanced the corresponding base model with jumping knowledge (JK) connections using JK-Concat [38] *without* changing the number of layers or other hyperparameters for the base method.

**Cora Full Benchmark Tuning**    The number of class labels in Cora-Full are many more compared to the other benchmarks (Table 5), which leads to a significant increase in the size of training parameters for each model. Therefore, we need to re-tune the hyperparameters, especially the regularization weights and learning rates, in order to get reasonable performance. We test the following command-line arguments for each baseline method:

- **$H_2$GCN-1 & $H_2$GCN-2**:
  - Dimension of Feature Embedding $p$: 64
  - Non-linearity Function $\sigma$: {ReLU, None}
  - Dropout Rate: $\{0, 0.5\}$
  - L2 Regularization Weight: {1e-5, 1e-6}
- **GCN** [17]:

- – `hidden1`: 64
- – `early_stopping`: $\{40, 100, 200\}$
- – `weight_decay`: $\{$5e-5, 1e-5, 1e-6$\}$
- – `epochs`: 2000

- **GCN-Cheby** [17]:
  - – `hidden1`: 64
  - – `weight_decay`: $\{$5e-5, 1e-5, 1e-6$\}$
  - – `max_degree`: 2
  - – `early_stopping`: $\{40, 100, 200\}$
  - – `epochs`: 2000

- **GraphSAGE** [11]:
  - – `hid_units`: 64
  - – `lr`: 0.7
  - – `epochs`: 2000

- **MixHop** [1]:
  - – `adj_pows`: 0, 1, 2
  - – `hidden_dims_csv`: $\{64, 192\}$
  - – `l2reg`: $\{$5e-4, 5e-5$\}$

- **GAT** [36]:
  - – `hid_units`: 8
  - – `l2_coef`: $\{$5e-4, 5e-5, 1e-5$\}$

- **MLP**
  - – Dimension of Feature Embedding $p$: 64
  - – Non-linearity Function $\sigma$: $\{$ReLU, None$\}$
  - – Dropout Rate: $\{0, 0.5\}$
  - – L2 Regularization Weight: 1e-5
  - – Learning Rate: 0.05

For **GCN+JK**, **GCN-Cheby+JK** and **GraphSAGE+JK**, we enhanced the corresponding base model with jumping knowledge (JK) connections using JK-Concat [38] *without* changing the number of layers or other hyperparameters for the base method.

# G   Synthetic Datasets: Details

## G.1   Data Generation Process & Setup

**Synthetic graph generation**   We generate synthetic graphs with various heterophily levels by adopting an approach similar to [1, 16]. In general, the synthetic graphs are generated by a modified preferential attachment process [3]: The number of class labels $|\mathcal{Y}|$ in the synthetic graph is prescribed. Then, starting from a small initial graph, new nodes are added into the graph one by one, until the number of nodes $|\mathcal{V}|$ has reached the preset level. The probability $p_{uv}$ for a newly added node $u$ in class $i$ to connect with an existing node $v$ in class $j$ is proportional to both the class compatibility $H_{ij}$ between class $i$ and $j$, and the degree $d_v$ of the existing node $v$. As a result, the degree distribution for the generated graphs follow a power law, and the heterophily can be controlled by class compatibility matrix $\mathbf{H}$. Table 3 shows an overview of these synthetic benchmarks, and more detailed statistics can be found in Table G.1.

**Node features & classes**   Nodes are assigned randomly to each class during the graph generation. Then, in each synthetic graph, the feature vectors of nodes in each class are generated by sampling feature vectors of nodes from the corresponding class in a real benchmark (e.g., Cora [30, 39] or `ogbn-products` [13]): We first establish a class mapping $\psi : \mathcal{Y}_s \rightarrow \mathcal{Y}_b$ between classes in the synthetic graph $\mathcal{Y}_s$ to classes in an existing benchmark $\mathcal{Y}_b$. The only requirement is that the class size in the existing benchmark is larger than that of the synthetic graph so that an injection between nodes from both classes can be established, and the feature vectors for the synthetic graph can be sampled accordingly. For `syn-products`, we further restrict the feature sampling to ensure that nodes in the training, validation and test splits are only mapped to nodes in the corresponding splits in the benchmark. This process respects the data splits used in `ogbn-products`, which are more realistic and challenging than random splits [13]. For simplicity, in our synthetic benchmarks, all the classes (5 for `syn-cora` and 10 for `syn-products` – Table G.1) are of the same size.

Table G.1: Statistics for Synthetic Datasets

| Benchmark Name | syn-cora | syn-products |
|---|---|---|
| # Nodes | 1490 | 10000 |
| # Edges | 2965 to 2968 | 59640 to 59648 |
| # Classes | 5 | 10 |
| Features | cora [30, 39] | ogbn-products [13] |
| Homophily $h$ | $[0, 0.1, \ldots, 1]$ | $[0, 0.1, \ldots, 1]$ |
| Degree Range | 1 to 94 | 1 to 336 |
| Average Degree | 3.98 | 11.93 |

**Experimental setup**   For each heterophily ratio $h$ of each benchmark, we independently generate 3 different graphs. For `syn-cora` and `syn-products`, we randomly partition 25% of nodes into training set, 25% into validation and 50% into test set. All methods share the same training, partition and test splits, and the average and standard derivation of the performance values under the 3 generated graphs are reported as the performance under each heterophily level of each benchmark.

## G.2   Detailed Results on Synthetic Benchmarks

Tables G.2 and G.3 give the results on `syn-cora` and `syn-products` shown in Figure 2 of the main paper (§ 5.1). Table G.4 provides the detailed results of the ablation studies that we designed in order to investigate the significance of our design choices, and complements Fig. 3 in § 5.1.

Table G.2: `syn-cora` (Fig. 2a): Mean accuracy and standard deviation per method and synthetic dataset (with different homophily ratio $h$). Best method highlighted in gray.

| h | 0.00 | 0.10 | 0.20 | 0.30 | 0.40 | 0.50 |
|---|---|---|---|---|---|---|
| **H$_2$GCN-1** | $77.40_{\pm0.89}$ | $76.82_{\pm1.30}$ | $73.38_{\pm0.95}$ | $75.26_{\pm0.56}$ | $75.66_{\pm2.19}$ | $80.22_{\pm1.35}$ |
| **H$_2$GCN-2** | $77.85_{\pm1.63}$ | $76.87_{\pm0.43}$ | $74.27_{\pm1.30}$ | $74.41_{\pm0.43}$ | $76.33_{\pm1.35}$ | $79.60_{\pm0.48}$ |
| **GraphSAGE** | $75.97_{\pm1.94}$ | $72.89_{\pm2.42}$ | $70.56_{\pm1.42}$ | $71.81_{\pm0.67}$ | $72.04_{\pm1.68}$ | $76.55_{\pm0.81}$ |
| **GCN-Cheby** | $74.23_{\pm0.54}$ | $68.10_{\pm1.75}$ | $64.70_{\pm1.17}$ | $66.71_{\pm1.63}$ | $68.14_{\pm1.56}$ | $73.33_{\pm2.05}$ |
| **MixHop** | $62.64_{\pm1.16}$ | $58.93_{\pm2.84}$ | $60.89_{\pm1.20}$ | $65.73_{\pm0.41}$ | $67.87_{\pm4.01}$ | $70.11_{\pm0.34}$ |
| **GCN** | $33.65_{\pm1.68}$ | $37.14_{\pm4.60}$ | $42.82_{\pm1.89}$ | $51.10_{\pm0.77}$ | $56.91_{\pm2.56}$ | $66.22_{\pm1.04}$ |
| **GAT** | $30.16_{\pm1.32}$ | $33.11_{\pm1.20}$ | $39.11_{\pm0.28}$ | $48.81_{\pm1.57}$ | $55.35_{\pm2.35}$ | $64.52_{\pm0.47}$ |
| **MLP** | $72.75_{\pm1.51}$ | $74.85_{\pm0.76}$ | $74.05_{\pm0.69}$ | $73.78_{\pm1.14}$ | $73.33_{\pm0.34}$ | $74.81_{\pm1.90}$ |

| h | 0.60 | 0.70 | 0.80 | 0.90 | 1.00 |
|---|---|---|---|---|---|
| **H$_2$GCN-1** | $83.62_{\pm0.82}$ | $88.14_{\pm0.31}$ | $91.63_{\pm0.77}$ | $95.53_{\pm0.61}$ | $99.06_{\pm0.27}$ |
| **H$_2$GCN-2** | $84.43_{\pm1.89}$ | $88.28_{\pm0.66}$ | $92.39_{\pm1.34}$ | $95.97_{\pm0.59}$ | $100.00_{\pm0.00}$ |
| **GraphSAGE** | $81.25_{\pm1.04}$ | $85.06_{\pm0.51}$ | $90.78_{\pm1.02}$ | $95.08_{\pm1.16}$ | $99.87_{\pm0.00}$ |
| **GCN-Cheby** | $78.88_{\pm0.21}$ | $84.92_{\pm1.03}$ | $90.92_{\pm1.62}$ | $95.97_{\pm1.07}$ | $100.00_{\pm0.00}$ |
| **MixHop** | $79.78_{\pm1.92}$ | $84.43_{\pm0.94}$ | $91.90_{\pm2.02}$ | $96.82_{\pm0.08}$ | $100.00_{\pm0.00}$ |
| **GCN** | $77.32_{\pm1.17}$ | $84.52_{\pm0.54}$ | $91.23_{\pm1.29}$ | $96.11_{\pm0.82}$ | $100.00_{\pm0.00}$ |
| **GAT** | $76.29_{\pm1.83}$ | $84.03_{\pm0.97}$ | $90.92_{\pm1.51}$ | $95.88_{\pm0.21}$ | $100.00_{\pm0.00}$ |
| **MLP** | $73.42_{\pm1.07}$ | $71.72_{\pm0.62}$ | $72.26_{\pm1.53}$ | $72.53_{\pm2.77}$ | $73.65_{\pm0.41}$ |

Table G.3: `syn-products` (Fig. 2b): Mean accuracy and standard deviation per method and synthetic dataset (with different homophily ratio $h$). Best method highlighted in gray.

| h | 0.00 | 0.10 | 0.20 | 0.30 | 0.40 | 0.50 |
|---|---|---|---|---|---|---|
| **H$_2$GCN-1** | $82.06_{\pm0.24}$ | $78.39_{\pm1.56}$ | $79.37_{\pm0.21}$ | $81.10_{\pm0.22}$ | $84.25_{\pm1.08}$ | $88.15_{\pm0.28}$ |
| **H$_2$GCN-2** | $83.37_{\pm0.38}$ | $80.03_{\pm0.84}$ | $81.09_{\pm0.41}$ | $82.79_{\pm0.49}$ | $86.73_{\pm0.66}$ | $90.75_{\pm0.43}$ |
| **GraphSAGE** | $77.66_{\pm0.72}$ | $74.04_{\pm1.07}$ | $75.29_{\pm0.82}$ | $76.39_{\pm0.24}$ | $80.49_{\pm0.96}$ | $84.51_{\pm0.51}$ |
| **GCN-Cheby** | $84.35_{\pm0.62}$ | $76.95_{\pm0.30}$ | $77.07_{\pm0.49}$ | $78.43_{\pm0.73}$ | $85.09_{\pm0.29}$ | $89.66_{\pm0.53}$ |
| **MixHop** | $15.39_{\pm1.38}$ | $11.91_{\pm1.17}$ | $14.03_{\pm1.70}$ | $14.92_{\pm0.56}$ | $17.04_{\pm0.40}$ | $18.90_{\pm1.49}$ |
| **GCN** | $56.44_{\pm0.59}$ | $51.51_{\pm0.56}$ | $54.97_{\pm0.66}$ | $64.90_{\pm0.90}$ | $76.25_{\pm0.04}$ | $86.43_{\pm0.58}$ |
| **GAT** | $27.39_{\pm2.47}$ | $21.49_{\pm2.25}$ | $37.27_{\pm3.99}$ | $44.46_{\pm0.68}$ | $51.86_{\pm8.52}$ | $69.42_{\pm5.30}$ |
| **MLP** | $68.63_{\pm0.58}$ | $68.20_{\pm1.20}$ | $68.85_{\pm0.73}$ | $68.65_{\pm0.18}$ | $68.37_{\pm0.85}$ | $68.70_{\pm0.61}$ |

| h | 0.60 | 0.70 | 0.80 | 0.90 | 1.00 |
|---|---|---|---|---|---|
| **H$_2$GCN-1** | $92.39_{\pm0.06}$ | $95.69_{\pm0.19}$ | $98.09_{\pm0.23}$ | $99.63_{\pm0.13}$ | $99.93_{\pm0.01}$ |
| **H$_2$GCN-2** | $94.81_{\pm0.27}$ | $97.67_{\pm0.18}$ | $99.13_{\pm0.05}$ | $99.89_{\pm0.01}$ | $99.99_{\pm0.01}$ |
| **GraphSAGE** | $89.51_{\pm0.29}$ | $93.61_{\pm0.52}$ | $96.66_{\pm0.19}$ | $98.78_{\pm0.11}$ | $99.63_{\pm0.08}$ |
| **GCN-Cheby** | $94.99_{\pm0.34}$ | $98.26_{\pm0.11}$ | $99.58_{\pm0.11}$ | $99.93_{\pm0.06}$ | $100.00_{\pm0.00}$ |
| **MixHop** | $19.47_{\pm5.21}$ | $21.15_{\pm2.28}$ | $24.16_{\pm3.19}$ | $23.21_{\pm5.30}$ | $25.09_{\pm5.08}$ |
| **GCN** | $93.35_{\pm0.28}$ | $97.61_{\pm0.24}$ | $99.33_{\pm0.08}$ | $99.93_{\pm0.01}$ | $99.99_{\pm0.01}$ |
| **GAT** | $85.36_{\pm3.67}$ | $93.52_{\pm1.93}$ | $98.84_{\pm0.12}$ | $99.87_{\pm0.06}$ | $99.98_{\pm0.02}$ |
| **MLP** | $68.21_{\pm0.93}$ | $68.72_{\pm1.11}$ | $68.10_{\pm0.54}$ | $68.36_{\pm1.42}$ | $69.08_{\pm1.03}$ |

Table G.4: Ablation studies of $H_2$GCN to show the significance of designs D1-D3 (Fig. 3(a)-(c)): Mean accuracy and standard deviation per method on the `syn-products` networks.

| Design | h | 0.00 | 0.10 | 0.20 | 0.30 | 0.40 | 0.50 |
|--------|---|------|------|------|------|------|------|
| D1-D3 | **[S0 / K2] H$_2$GCN-1** | 82.06±0.24 | 78.39±1.56 | 79.37±0.21 | 81.10±0.22 | 84.25±1.08 | 88.15±0.28 |
| D3 | **H$_2$GCN-2** | 83.37±0.38 | 80.03±0.84 | 81.09±0.41 | 82.79±0.49 | 86.73±0.66 | 90.75±0.43 |
| D1 | **[NS0] N$_1$ + N$_2$** | 52.72±0.13 | 41.65±0.18 | 46.11±0.86 | 58.16±0.79 | 71.10±0.54 | 82.19±0.40 |
| D1 | **[NS1] Only N$_1$** | 40.35±0.58 | 35.17±0.92 | 40.35±0.92 | 52.45±0.85 | 65.62±0.56 | 76.05±0.38 |
| D1, D2 | **[S1 / N2] w/o $\bar{N}_2$** | 79.65±0.27 | 76.08±0.76 | 76.46±0.21 | 77.29±0.46 | 79.81±0.88 | 83.56±0.22 |
| D2 | **[N1] w/o $\bar{N}_1$** | 72.27±0.55 | 73.05±1.23 | 75.81±0.67 | 76.83±0.72 | 80.49±0.72 | 82.91±0.44 |
| D2 | **[N0] w/o 0-hop neighb. (ego)** | 63.55±0.46 | 46.73±0.42 | 42.29±0.55 | 48.20±0.59 | 61.22±0.35 | 75.15±0.27 |
| D3 | **[K0] No Round-0** | 75.63±0.19 | 61.99±0.57 | 56.36±0.56 | 61.27±0.71 | 73.33±0.88 | 84.51±0.50 |
| D3 | **[K1] No Round-1** | 75.75±0.90 | 75.65±0.73 | 79.25±0.18 | 81.19±0.33 | 84.64±0.35 | 88.46±0.60 |
| D3 | **[R2] Only Round-2** | 73.11±1.01 | 62.47±1.35 | 59.99±0.43 | 64.37±1.14 | 75.43±0.70 | 86.02±0.79 |
| § D.2 | **Non-linear H$_2$GCN-2 (§ D.2)** | 82.23±0.25 | 78.78±1.04 | 80.47±0.15 | 82.08±0.10 | 85.89±0.53 | 89.78±0.11 |

| Design | h | 0.60 | 0.70 | 0.80 | 0.90 | 0.99 | 1.00 |
|--------|---|------|------|------|------|------|------|
| D1, D3 | **[S0 / K2] H$_2$GCN-1** | 92.39±0.06 | 95.69±0.19 | 98.09±0.23 | 99.63±0.13 | 99.88±0.06 | 99.93±0.01 |
| D3 | **H$_2$GCN-2** | 94.81±0.27 | 97.67±0.18 | 99.13±0.05 | 99.89±0.08 | 99.98±0.00 | 99.99±0.01 |
| D1 | **[NS0] N$_1$ + N$_2$** | 90.39±0.54 | 95.25±0.06 | 98.27±0.13 | 99.69±0.03 | 99.98±0.02 | 100.00±0.00 |
| D1 | **[NS1] Only N$_1$** | 84.41±0.44 | 90.15±0.27 | 95.21±0.34 | 97.71±0.06 | 99.56±0.11 | 99.49±0.11 |
| D1, D2 | **[S1 / N2] w/o $\bar{N}_2$** | 87.39±0.33 | 91.08±0.50 | 94.36±0.32 | 97.01±0.40 | 98.79±0.23 | 98.71±0.15 |
| D2 | **[N1] w/o $\bar{N}_1$** | 87.24±0.21 | 92.55±0.50 | 95.64±0.19 | 98.71±0.13 | 99.73±0.12 | 99.83±0.06 |
| D2 | **[N0] w/o 0-hop neighb. (ego)** | 86.08±0.58 | 93.03±0.29 | 97.45±0.09 | 99.45±0.06 | 99.98±0.02 | 99.98±0.03 |
| D3 | **[K0] No Round-0** | 92.42±0.13 | 96.81±0.11 | 99.09±0.27 | 99.89±0.01 | 100.00±0.00 | 100.00±0.00 |
| D3 | **[K1] No Round-1** | 93.05±0.23 | 97.17±0.36 | 99.06±0.09 | 99.89±0.08 | 99.97±0.02 | 99.97±0.01 |
| D3 | **[R2] Only Round-2** | 93.79±0.28 | 97.88±0.18 | 99.38±0.12 | 99.89±0.05 | 100.00±0.00 | 100.00±0.00 |
| § D.2 | **Non-linear H$_2$GCN-2** | 93.68±0.50 | 96.73±0.23 | 98.55±0.06 | 99.74±0.05 | 99.96±0.04 | 99.93±0.03 |

# H  Real Datasets: Details

**Datasets**   In our experiments, we use the following real-world datasets with varying levels of homophily ratios $h$. Some network statistics are given in Table 5.

- **Texas, Wisconsin and Cornell** are graphs representing links between web pages of the corresponding universities, originally collected by the CMU WebKB project. We used the preprocessed version in [26]. In these networks, nodes are web pages, which are classified into 5 categories: course, faculty, student, project, staff.

- **Squirrel and Chameleon** are subgraphs of web pages in Wikipedia discussing the corresponding topics, collected by [29]. For the classification task, we utilize the class labels generated by [26], where the nodes are categorized into 5 classes based on the amount of their average traffic.

- **Actor** is a graph representing actor co-occurrence in Wikipedia pages, processed by [26] based on the film-director-actor-writer network in [35]. We also use the class labels generated by [26].

- **Cora, Pubmed and Citeseer** are citation graphs originally introduced in [30, 22], which are among the most widely used benchmarks for semi-supervised node classification [31, 13]. Each node is assigned a class label based on the research field. These datasets use a bag of words representation as the feature vector for each node.

- **Cora Full** is an extended version of Cora, introduced in [4, 31], which contain more papers and research fields than Cora. This dataset also uses a bag of words representation as the feature vector for each node.

**Data Limitations**   As discussed in [31, 13], Cora, Pubmed and Citeseer are widely adopted as benchmarks for semi-supervised node classification tasks; however, all these benchmark graphs display strong homophily, with edge homophily ratio $h \geq 0.7$. As a result, the wide adaptation of these benchmarks have masked the limitations of the homophily assumption in many existing GNN models. Open Graph Benchmark is a recent effort of proposing more challenging, realistic benchmarks with improved data quality comparing to the existing benchmarks [13]. However, with respect to homophily, we found that the proposed OGB datasets display homophily $h > 0.5$.

In our synthetic experiments (§ G), we used `ogbn-products` from this effort to generate higher quality synthetic benchmarks while varying the homophily ratio $h$. In our experiments on real datasets, we go beyond the typically-used benchmarks (Cora, Pubmed, Citeseer) and consider benchmarks with strong heterophily (Table 5). That said, these datasets also have limitations, including relatively small sizes (e.g., WebKB benchmarks), artificial classes (e.g., Squirrel and Chameleon have class labels based on ranking of page traffic), or unusual network structure (e.g., `Squirrel` and `Chameleon` are dense, with many nodes sharing the same neighbors — cf. § 5.2). We hope that this paper will encourage future work on more diverse datasets with different levels of homophily, and lead to higher quality datasets for benchmarking GNN models in the heterophily settings.