[Reviews · NeurIPS 2020]

Review 1

Summary and Contributions: The submission studies the problem of graph neural networks. The goal is to improve the representation power of graph neural networks for graphs that do not follow the networking principle of homophily. The authors argue that existing GNN models only focus on modeling structural homophily in graphs. To make GNNs expressive for graphs beyond homophily, the authors introduce a GNN model, H2GNN, with three new and minor model designs : ego- and neighbor-embedding separation, higher-order neighborhoods, and combination of intermediate representations. The empirical demonstration is conducted on both synthetic datasets and real world datasets. The results suggest that the proposed designs work well on synthetic graphs as expected while it can not consistently outperform existing (simple and outdated) baselines with most cases having marginal improvements.

Strengths: The motivations to improve graph neural networks’ representation capacity is interesting. Certainly the problem of graph neural networks itself is important, given its promising applications for many fields. The presentation of this submission is nice: well written, organized, and flowed. It is easy to follow and understand the arguments.

Weaknesses: The major concerns of this work lie in two folds: 1) technical contributions for graph neural net research, and 2) the experimental design and results. The authors argue that we need to enable graph neural nets to model graphs beyond homophily, which is reasonable and great. However, the three corresponding designs that are introduced to address this issue lack of technical novelty and depth. All of the three designs have been proposed and well utilized (in a separated way) in existing graph neural nets. The proposed H2GNN model puts all three design together without clear discussions about their original sources during the authors’ arguments (though table 2 is used in related work). Furthermore, the goal of the three designs is to model heterophily in graphs or networks. It is not clear how the second and third designs help with this goal. [[ In nature, graph neural nets of multiple layers are able to model the network heterophily.---the author's response on this point is convincing, so plz ignore this sentence]] Not sure how these methods specifically benefit graph neural nets for greater modeling power of heterophily. In other words, it would be straightforward to see how different layers of graph neural nets help with the issue that the authors are interested in. Putting together, it is difficult to consider the direct usage of three existing techniques together advance the graph neural network research. The other concern is on its experiments, both the design and results. a) Three designs are proposed to improve graph neural nets, it is straightforward and critically important to demonstrate how each of them helps with the performance, that is, the modeling of network heterophily. This is particularly necessary given the overall performance of the proposed model is quite limited with nearly half of the cases (4/10) weaker than simple baselines and the remaining cases (6/10) getting only marginal improvements. b) On the three commonly used datasets for evaluating graph neural nets, core, citeseer, and pubmed, the performance of the proposed method is weak and simple methods achieve great results. More importantly, it is also difficult to see the performance differences between all models and it is pretty clear that simple models (GCN, Graphsage, GAT) do good enough jobs, making the complicated model design and combination convincingly unnecessary. c) These three datasets have been commonly used for evaluating graph neural nets (as a community we should improve this). Popular data splits are ignored here and instead different splitting strategies are used, as a result, the reported results are inconsistent with most literature. It would be nice if the common evaluation pipeline was adopted here for an easy and fair comparison. In addition, only several gnn baselines are considered here, and most of them are outdated and more strong baselines could be compared against. Overall, the technical contribution of this work is limited given all three proposed components have been well utilized in existing research, and the empirical results are not convincing as the experimental design requires significant improvements and the results are not as promising as the arguments.

Correctness: The method is correct, and the claims and empirical methodology might need further calibrations. Details see above.

Clarity: The paper is well written and easy to follow.

Relation to Prior Work: Partially. The proposed model introduces three components for graph neural nets, while all of them have been well studied and utilized in existing graph neural nets. During the introduction to each of them, the connection with previous contributions is not clearly discussed.

Reproducibility: No

Additional Feedback:


Review 2

Summary and Contributions: This paper provides novel designs for graph neural networks, which are usually based on a homophily assumption, to increase representational power to learn heterophily or dissimilar features as well. The authors demonstrate intuition/theoretical justification/observations for each design to support the proposed model and show that the proposed model H2GCN with the proposed 3 different designs outperforms all other baselines on semi-supervised node classification tasks.

Strengths: First, the paper is well-motivated and the proposed idea is well-designed to overcome the known limitations of existing graph neural networks, homophily assumption. While the homophily assumption seems reasonable (e.g., similar nodes are connected to each other), it is not always true and for heterophily graphs, existing models extremely struggle to classify nodes correctly. Thus, it is necessary to find a model which is able to handle the dissimilar features properly. Second, the proposed designs are well-explained with appropriate intuition/justification/observations. Moreover, related theorems support the necessity of the proposed designs. Third, the authors rigorously evaluate the proposed design over many datasets and investigate the significance of design choices. I believe that people attending NeurIPS are interested in the experimental results. Finally, the theoretical analysis provided in the supplementary material will be beneficial for understanding the behavior of the proposed model.

Weaknesses: Overall, there are not many weaknesses in the paper, however, I have some questions about the novelty of this work in the additional feedback part.

Correctness: To the best of my knowledge, the claims and method are correct. The model’s architecture/intuition/justification/experimental setting are correct.

Clarity: Yes, this paper is well written and clear.

Relation to Prior Work: It is clearly explained which existing works have targeted on similar tasks and how this work differs from the previous works.

Reproducibility: Yes

Additional Feedback: First of all, I fully agree with the motivation, heterophily can be an important feature to correctly classify nodes in a graph. As many existing works aggregate neighboring representations with an ego-representation by mean-pooling, the dissimilarity between the ego-node and its neighbors is diluted. Thus, it is necessary to consider ego and neighboring nodes separately. While this paper handles the heterophily properly, I wonder if this work really provides a new representation which existing models are not able to extract. The main idea behind of the three proposed designs can be summarized as follows: “Let’s separate nodes in sub-groups as much as possible and keep each group’s representation as much as possible” In other words, instead of mixing (or merging) two nodes which are separable by some criteria, handle the nodes separately as much as possible. The separation rules are (1) the number of hops (D1, D2) and (2) the number of layers (D3). I believe that the main idea I described above is actually not directly related to heterophily but it is more related to how to maximize the expressiveness of graph neural networks. So, the logic flow might be “we increase the representation power of gnn as much as possible. As a result, our model becomes able to handle heterophily nodes successfully.” And this is why other baseline GraphSAGE and GCN-Cheby, which are more expressive than GCN, show similar accuracy curves in Figure 2. So, my question is if the true difference between H2GCN and GCN is “concatenate(input nodes) vs. sum(or mean)(input nodes)”. And my second question is where the performance difference between two concatenate models (H2GCN vs GraphSAGE) comes from. D2 and D3 might be helpful since it doesn’t lose information from N_2(v) and intermediate layers, respectively. However, because of the concatenated embeddings, the number of learnable parameters in H2GCN should be significantly increased if other settings are same (According to the supplementary material, the hidden dimension of H2GCN and other baselines are similar) and the performance gain might come from the increased parameters rather than the proposed design. Still, I think that it is interesting to see how the heterophily is handled by the concatenated embeddings and it gives me an insight about designing graph neural networks. However, it would be much better to provide how many learnable parameters are used in H2GCN and how its performance is affected by the number of parameters. *** Thanks for the detailed response. After carefully reading the authors response, I agree that this work is more about studying effectiveness in heterophily settings with in-depth theoretical justifications rather than introducing (so called) new architectural model. I think that this paper aims to deliver some meaningful analysis about heterophily, which is not thoroughly studied (to the best of my knowledge) so far. (I understand that existing works can handle the heterophily on graphs but they didn’t study the heterophily or low homophily properties explicitly.) However, considering other aspects (architectural novelty, experimental results), I would like lower the score to 6.


Review 3

Summary and Contributions: This work investigates the use of graph neural networks (GNNs) for semi-supervised node classification datasets in the context of heterophilic (as opposed to homophilous) graph data (i.e., cases where node labels tend to differ between connected nodes). The paper highlights the fact that most popular GNN architectures naturally have inductive biases towards homophily and are not well-suited for data exhibiting heterophily. Both theoretical and empirical statements are made to substantiate this idea, including claims regarding the limitations of some popular GNN variants (e.g., basic GCNs). The paper then proposes three important design decisions to make GNNs effective on heterophilic data: 1) separating node and neighbor embeddings during learning, 2) aggregating information from different hops, and 3) using jumping knowledge (JK) connections. Experiments on both real and synthetic datasets support the utility of the proposed architecture changes.

Strengths: - The problem is well-motivated. As the paper points out, most GNN works implicitly assume that the data is homophilous, so there is value in highlighting the prevalence and importance of non-homophilous data. - The paper is well-written, with a good flow of intuitive descriptions, theoretical statements, and empirical evidence. - The model decisions are motivated by theoretical statements. Rather than simply proposing ideas that seems intuitively reasonable, the architecture choices are supported by brief theoretical statements. This grounds the methodology and provides more rigor to the proposed approach. - The key trends (e.g., the utility of using JK-connections and higher-order neighbor features) will be useful to other researchers in the field working on non-homophilous data.

Weaknesses: * The authors response gives a satisfactory response to some of my main concerns below. In particular, the authors agree to clarify the key aspect of their novelty (i.e., the focus on heterophily) and have included the additional baselines I requested (e.g., GraphSAGE+JK connections). - One key shortcoming of the paper is in the methodological novelty. None of the proposed design decisions are novel. In particular, the idea of concatenating the ego embedding and the neighbor embedding was first proposed in GraphSAGE, the idea of leverage multiple different hops of neighborhoods has been employed in numerous recent works, most prominently MixHop, and the idea of concatenating the intermediate representations from the GNN layers was proposed in the JK paper. Indeed, the paper does not hide this fact (and all these prior works are properly cited), but it seems somewhat disingenuous to claim the proposed approach as a new kind of GNN when it is essentially a combination of three different modifications/modules proposed in prior work. - A second key shortcoming in the paper is in the thoroughness of the empirical results. Most prominently, there are not strong enough baselines. For example, the work compares against some standard GNN variants (e.g., GCNs and GraphSAGE), but none of these baselines are augmented with JK connections, which is a standard technique. It seems very unfair and misleading to only include baselines without JK connections, when JK connections are a standard technique that (a) was proposed as a generic plug-in module that can be combined with most GNNs, (b) can be and has been combined with those baselines by other works (e.g., in the original JK paper) and (c) is used in the proposed approach. Indeed, a variant of MixHop with JK connections would essentially be the same algorithm as the one proposed in this work. This also relates to the lack of ablation studies on the real-world data. For example, is it the MixHop-style aggregation over different neighborhoods or JK connections that are leading to the biggest performance improvements on the real-world datasets? The lack of these stronger baselines is especially troubling given that the performance improvements are relatively marginal compared to GraphSAGE. It seems to be a reasonable guess that GraphSAGE+JK connections would be competitive the proposed approach, so these comparisons need to be included. - In addition to the thoroughness of the empirical results, a weakness of the paper is the lack of significance and stability in the results. In many of the datasets, the performance differences between the different approaches seem to be within the confidence intervals (based on the reported standard deviations). Some more thorough statistical tests or more experiments would improve the paper to give a stronger sense of which differences are actually significant. - Some of the theorems involve "straw man" arguments, where limitations of very simple/impoverished GNN variants are highlighted. Most prominently, Theorem 1 highlights a limitation of GCN-style models that use a self-loop, but this theorem does not apply for many GNNs (e.g. GraphSAGE). It seems unfair to imply that Theorem 1 is pointing out a limitation of GNNs, when it is actually pointing out a limitation of one specific (though popular) variant.

Correctness: * The authors acknowledge and satisfactorily responded to the issue raised below in their response. The statement that "the adjacency matrix is a low-pass filter" is incorrect. In fact, high-order polynomials of the (normalized) adjacency matrix would correspond to low-pass filters. Indeed, a canonical "low-pass" filter defined using a normalized adjacency matrix would correspond to the fixed point of the recurrence $x^t = D^{-1}Ax^{t-1}$ (i.e. $(D^{-1}A)^n x$ for a sufficiently large n), which would essentially correspond to the asymptotic state of a simple diffusion process on the graph. In this steady state, we would have that signals are "smooth" (i.e., near constant) within connected components on the graph, which is maximally low-pass in the sense that this filter would correspond to the eigenvector(s) with zero eigenvalue(s) of the analogously normalized Laplacian. Thus, low-pass filters really correspond to high powers of the (normalized) adjacency matrix, and it is incorrect to say that aggregating over larger neighborhood sizes makes the filters more sensitive to "high frequencies".

Clarity: Overall, the paper is relatively clear. It is somewhat dense and sparse on detail in the main text, but this is more a reflection of the amount of content, rather than issues in the writing. The grammar and polish is acceptable, though there are some typographical errors (e.g., spacing issues after equations, unnecessary commas after em-dashes).

Relation to Prior Work: A sufficient number of related works are cited, though the formatting in the references section could be improved (e.g., consistent formatting of conference names). However, the paper does not properly engage with all these related works; in particular, the paper does not clearly explain the methodological novelty w.r.t. these previous works. This issue is also discussed in the review section on the "weaknesses" of the paper, but the key problem is that the paper does not clearly explain if (or how) their combination of the three design decisions (D1-D3) is a "new model". For example, in the section D3, the paper does state that "this design is introduced in jumping knowledge networks", but then the section goes on to intuitions and justifications for this component, as if it were a novel contribution. Moreover, the paper seems to imply that the use of JK connections in combination with higher-order neighborhood information is a new model, but this is a somewhat tenuous claim, since the JK approach was proposed as a general plug-and-play module to be combined with arbitrary GNN architectures, and using higher-order neighborhood information has been a popular approach in recent years. Essentially, the paper would benefit from a much more clear and focused discussion of how the H_2GCN model is "new", as opposed to just being a straightforward application of existing techniques.

Reproducibility: Yes

Additional Feedback: Overall, I enjoyed reading this paper, but there are significant improvements that could be made. Most prominently, the experiments section should be reframed in such a way that improvements from D1-D3 are separately evaluated on the real-world data, with a fair treatment of baselines (e.g., including GraphSAGE+JK and MixHop+JK as baselines). In this way, the paper can focus on evaluating the importance of D1-D3, rather than proposing a "new model" (which largely combines existing techniques) and making inherently unfair comparisons to baselines (by not allowing them to use existing designs that the "new model" employs). Indeed, the paper in its current state seems to obsfucate the relationship to prior work in order to claim novelty. For instance, in lines 318-319, the paper claims that only using the output of the final layer of message passing is "akin to what other GNNs do". However, this is not the case, as the JK-connection paper explicitly contributes an approach that proposes to use representations from different layers and that can be combined with most GNNs. Again, the paper often implicitly claims D1-D3 as contributions of this work and comparisons to baselines and "other GNNs" do not have access to these design improvements, even though all these improvements have been previously proposed. In my view, this work could be really strong if it avoided this obsfucation and was more forthright about the relation to prior work. I.e., there is no "new model" proposed. Instead, this work contributes a thorough investigation of what GNN variants are most effective for learning on non-homophilous data.


Review 4

Summary and Contributions: The existing GNNs mostly deal with graphs with strong homophily and fail to handle graphs with heterophily or low homophily. This submission proposes to tackle the issue by (1) convoluting not only the first-order but also the second-order neighbors, and (2) combining the representations produced by the intermedia layers (as in Jumping Knowledge Networks). The solution is theoretically justified.

Strengths: 1. It addresses a vital limitation of the existing GNNs. 2. Good theoretical analyses. 3. An easy-to-implement solution that is both intuitively and theoretically sound.

Weaknesses: 1. Only results on small datasets. 2. Second-order neighbors may bring some technical challenges (e.g., high variance) when one attempts to scale the proposed method to large-scale graphs via neighborhood sampling.

Correctness: Yes.

Clarity: Yes.

Relation to Prior Work: Yes.

Reproducibility: Yes

Additional Feedback:

[Author Response · NeurIPS 2020]

We thank the reviewers for their thoughtful and constructive feedback. We are pleased that all reviewers [**R1**, **R2**,
**R3**, **R4**] find the paper clear; most reviewers [**R2**, **R3**, **R4**] find the problem of overcoming the implicit homophily
assumption in most GNN models well-motivated and vital; and [**R2**, **R3**, **R4**] value our theoretical analysis and the
grounding of our methodology. Next, we first clarify the technical contributions of our work, and then address specific
comments. While we only address major discussion points here, we will incorporate all feedback in the final version.

**Recap: Technical contributions & Novelty.** We empirically revealed the limitation of some widely-adopted GNNs
to learn over *networks with heterophily*, and *identified* a set of key designs that actually are helpful. We showed the
effectiveness of these designs under heterophily through theoretical analysis (§3.1.1 - 3.1.3) and ablation studies (§5.1,
lines 298–321). While we acknowledge that these designs are used in existing methods, we are the first to revisit their
effectiveness *in heterophily settings with in-depth theoretical justifications and extensive empirical evaluation* (this has
been largely unknown before this work). Existing models have used subsets of these designs (and tested them under
strong homophily), but not all at the same time (Table 2). Thus, our purpose in designing $H_2$GCN is to exemplify how
an effective combination of these designs can help a GNN better adapt to the whole spectrum of low-to-high homophily,
while avoiding interference with other designs. We'll revise our paper to clarify the scope of our contributions.

[**R1**, **R3**] **Concerns that the proposed designs aren't novel as they're existing techniques.** We are the first to
discuss the importance of these designs *under heterophily* with novel theoretical justifications and extensive empirical
evaluations. While we agree that the designs are not *new*, our analysis for the heterophily setting is *novel*. We believe
that showing *what* works and *why* in a challenging, rarely-studied setting advances the field. We'll make this more clear.

[**R1**, **R3**] **Sufficiency of baselines.** Thanks to the identified designs, we were able to spot very competitive baselines
under heterophily (e.g. GCN-Cheby, GraphSAGE), which were not compared against in recent state-of-the-art works
(e.g. GeomGCN [20] in ICLR'20, against which we also compare). We have put considerable effort in ensuring an
extensive, rigorous comparison. That said, we appreciate **R3**'s excellent suggestion to enhance the baselines with
the jumping-knowledge (JK) connections, corresponding to design D3. We use JK-Concat [34] and report results for
GraphSAGE, GCN-Cheby and GCN in Table R1. JK connections improve the baselines (for fixed number of layers) in
some cases though without changing our observations. We'll discuss these results in detail in the final version.

Table R1: [**R3**] Additional baselines on real benchmarks (baselines + JK). Our observations remain largely the same.

| | Texas | Wisconsin | Actor | Squirrel | Chameleon | Cornell | Cora Full | Citeseer | Pubmed | Cora |
|---|---|---|---|---|---|---|---|---|---|---|
| **GraphSAGE+JK** | $81.89_{\pm7.32}$ | $83.14_{\pm4.45}$ | $34.35_{\pm0.67}$ | $40.84_{\pm1.54}$ | $58.09_{\pm1.92}$ | $77.03_{\pm4.08}$ | $65.31_{\pm0.99}$ | $75.91_{\pm1.09}$ | $88.34_{\pm0.47}$ | $86.24_{\pm1.21}$ |
| **GCN-Cheby+JK** | $77.03_{\pm7.88}$ | $81.18_{\pm4.55}$ | $34.70_{\pm1.05}$ | $40.90_{\pm2.78}$ | $59.91_{\pm2.28}$ | $71.62_{\pm9.47}$ | $66.09_{\pm0.12}$ | $74.19_{\pm1.69}$ | $88.69_{\pm0.49}$ | $84.91_{\pm1.98}$ |
| **GCN+JK** | $66.49_{\pm6.64}$ | $74.31_{\pm6.43}$ | $34.26_{\pm0.90}$ | $39.43_{\pm1.00}$ | $62.70_{\pm1.98}$ | $64.59_{\pm8.68}$ | $64.73_{\pm0.30}$ | $74.53_{\pm1.60}$ | $88.45_{\pm0.49}$ | $85.81_{\pm1.04}$ |

[**R1**, **R3**] **Significance / stability of results on real data.** These benchmarks show the *complexity* of learning from
graphs with heterophily. Our main focus is *not* to optimize for high-homophily datasets like Cora, Citeseer, Pubmed
[**R1**]; we include them to show the trends across the full spectrum of low-to-high homophily. While we agree that
there is not a consistent winner for *all* the datasets, we have demonstrated that $H_2$GCN variants have the best *overall*
performance across the spectrum in terms of the smallest average ranking. Another clear trend is that most models
utilizing some of our identified (heterophily-friendly) designs outperform other models under heterophily; deviations
are related to implementation details and other designs that may interfere with our identified designs.

[**R1**] **"Graph neural nets of multiple layers are able to model the network heterophily."** This is not the case:
2-layer GCN performs poorly under heterophily (cf. Table 4) and in general can suffer from oversmoothing[1]. **"Not**
**clear how designs D2+D3 help in heterophily."** Removal of designs D2 and D3 leads to dramatic decrease in accuracy
under heterophily, as shown in the ablation studies in Fig. 3(b)-3(c) (§ 5.1; theoretical justifications in §3.1.2-3.1.3).

[**R2**] **Differences between $H_2$GCN and baselines**. We discuss the differences from GCN in lines 639–647, and from
GraphSAGE in lines 653–659 (Supp. §D.2). GraphSAGE generally has more learnable parameters than $H_2$GCN—e.g.,
$H_2$GCN-2 outperforms GraphSAGE in `syn-products` with less than $\frac{1}{5}$ of the parameters (10,880 vs. 59,648).

[**R3**] **"Thm 1 only points out a limitation of one specific (though popular) GCN variant."** This in fact illustrates
the point of our work: there exist GNNs that happen to make the design choices we study, but also popular GNNs that
do not. Without work to shed light on *why* GNNs should use particular designs, any success on heterophily is the result
of a shot in the dark. **"The adj matrix is a low-pass filter"** & **"Aggregation over larger neighborhood sizes makes**
**the filters more sensitive to high frequencies"** are incorrect. We agree that "high-order polynomials of the norm adj
matrix correspond to low-pass filter" is more accurate; we'll reword this. However, we have not found the latter claim
in our work. In lines 198–200 we say: "*intermediate outputs from earlier rounds contain higher-frequency components*
*than ... later rounds*"; thus, D3 helps when higher-frequency information is beneficial (e.g., in heterophily).

[**R4**] **"Small datasets ... technical challenges (e.g., high variance) when one attempts to scale the proposed**
**method via neighborhood sampling"** These are important future directions. Our paper calls for future work in
designing large-scale benchmarks exhibiting heterophily, which will hopefully inspire methodological developments.

## Footnotes

[1] Graph Neural Networks Exponentially Lose Expressive Power For Node Classification. ICLR 2020


[Meta-Review · NeurIPS 2020]

This paper discusses some implicit limitations of GNNs in terms of bias towards homophilous predictions. The paper presents three GNN architectural guidelines for combating this, which can lead to improved predictions, particularly on networks exhibiting heterophilous structure (i.e., non-homophilous labels). The design choices are motivated theoretically and intuitively and then combined into a single model that can provide better predictions on networks with heterophilous structure, as demonstrated by synthetic and real-world data experiments. The paper provides a number of interesting insights into why certain GNN architectural choices can help predictions in the case of low network homophily. Although not mentioned in their paper, a similar idea to higher-order neighborhoods (Section 3.1.2) has recently been deployed within more classical graph-based semi-supervised algorithms (i.e., label propagation or label smoothing) for predicting gender on Facebook networks, where gender is not majorly homophilous (see Altenburger-Ugander, Nature Human Behavior 2018 and Chin et al., WWW 2019; full references below). I believe that these ideas provide further motivation for the design choices appearing in this paper and including them will strengthen some of the intuition. Gender prediction is another task that the authors might consider. As pointed out by other reviewers, some of the methodological ideas already appear in the literature in a few places (see also my above comments). However, there is still important novelty in this paper, which was clarified in the authors’ response and subsequent discussion. Specifically, the central problem of making predictions in networks with heterophilous label structure is the motivator for the components, which is different from the original papers. Furthermore, there is clear theoretical and intuitive justification in the paper for why these approaches should help address the central problem. Because of this, I believe that the paper actually generates insight into why some existing ideas may work as well as they do while offering a new way to think about evaluation of GNNs. In addition, the setup of the experiments and evaluation (both on synthetic and real datasets) are novel. There is one dissenting opinion in the set of four reviewers. Reviewer #1’s concerns boil down to the following: (a) a lack of motivation for the methods; (b) the proposed methods sometimes do not perform as well as other ones; and (c) some missing benchmarks. These are all non-issues in my view. For (a), there is clear motivation in the paper, and this was listed as a strength by the three other reviewers. Issue (b) is a misunderstanding. The authors intentionally present some datasets with high levels of homophily where the proposed methods are not supposed to be better than existing approaches; this appeared in the author response (lines 26--32). Finally, the supposedly missing benchmarks in issue (c) (Cora, CiteSeer, and PubMed with standard train/validation/test splits) are actually in the paper. I think confusion arose because the paper includes one dataset (“Cora Full”, which has different training/validation/test splits) for which they did not have prediction performance of two baseline methods (GAT* and GEOM-GCN*), as the corresponding paper did not run experiments on that dataset. I commend the authors for addressing an interesting and important problem with intuition, theory, and experiments. The paper also opens the door to further research addressing the role of homophily or heterophily when making structured predictions. Some recent research has been circling these ideas (see below for a few more references), so I also think the research is timely. Overall, I would be very pleased to see this paper appear at NeurIPS. For the camera version, here are several recommendations that are in addition to smaller points appearing in the reviews. 1. Give proper credit for some of the individual components of the model (which have appeared elsewhere) and focus the paper on the problem (going beyond homophily). 2. Include a better description of the predictions. The non-homophilous attributes are not mentioned in the main text, even though they are described in the appendix. 3. Include some acknowledgment, discussion, and (possibly) appropriate comparison against other graph-based semi-supervised learning methods from the data mining community that deal with homophily / heterophily: -- Gatterbauer. Semi-Supervised Learning with Heterophily. 2014. -- Peel. Graph-based semi-supervised learning for relational networks. SDM, 2017. -- Altenburger and Ugander. Monophily in social networks introduces similarity among friends-of-friends. Nature Human Behavior, 2018. -- Chin et al. Decoupled smoothing on graphs. WWW, 2019. -- Jia and Benson. Residual Correlation in Graph Neural Network Regression. KDD, 2020.